# Endogenous oxytocin exerts anti-nociceptive and anti-inflammatory effects in rats

Haruki Nishimura[1,2,10], Mitsuhiro Yoshimura [1,3,10,11✉], Makiko Shimizu[1], Kenya Sanada[1], Satomi Sonoda[4], Kazuaki Nishimura[1], Kazuhiko Baba[1,2], Naofumi Ikeda[1,2], Yasuhito Motojima[2], Takashi Maruyama[1], Yuki Nonaka[1], Ryoko Baba[5], Tatsushi Onaka [6], Takafumi Horishita[7], Hiroyuki Morimoto[5], Yasuhiro Yoshida[8], Makoto Kawasaki [2], Akinori Sakai[2], Masafumi Muratani [9], Becky Conway-Campbell [3], Stafford Lightman [3] & Yoichi Ueta [1,11✉]

Oxytocin is involved in pain transmission, although the detailed mechanism is not fully understood. Here, we generate a transgenic rat line that expresses human muscarinic acetylcholine receptors (hM3Dq) and mCherry in oxytocin neurons. We report that clozapine-N-oxide (CNO) treatment of our oxytocin-hM3Dq-mCherry rats exclusively activates oxytocin neurons within the supraoptic and paraventricular nuclei, leading to activation of neurons in the locus coeruleus (LC) and dorsal raphe nucleus (DR), and differential gene expression in GABA-ergic neurons in the L5 spinal dorsal horn. Hyperalgesia, which is robustly exacerbated in experimental pain models, is significantly attenuated after CNO injection. The analgesic effects of CNO are ablated by co-treatment with oxytocin receptor antagonist. Endogenous oxytocin also exerts anti-inflammatory effects via activation of the hypothalamus-pituitary-adrenal axis. Moreover, inhibition of mast cell degranulation is found to be involved in the response. Taken together, our results suggest that oxytocin may exert anti-nociceptive and anti-inflammatory effects via both neuronal and humoral pathways.

[1] Department of Physiology, School of Medicine, University of Occupational and Environmental Health, Kitakyushu, Japan. [2] Department of Orthopaedic Surgery, School of Medicine, University of Occupational and Environmental Health, Kitakyushu, Japan. [3] Translational Health Sciences, Bristol Medical School, University of Bristol, Bristol, UK. [4] The First Department of Internal Medicine, School of Medicine, University of Occupational and Environmental Health, Kitakyushu, Japan. [5] Department of Anatomy (II), School of Medicine, University of Occupational and Environmental Health, Kitakyushu, Japan. [6] Division of Brain and Neurophysiology, Department of Physiology, Jichi Medical University, Shimotsuke, Japan. [7] Department of Anesthesiology, School of Medicine, University of Occupational and Environmental Health, Kitakyushu, Japan. [8] Department of Immunology and Parasitology, School of Medicine, University of Occupational and Environmental Health, Kitakyushu, Japan. [9] Genome Biology, Faculty of Medicine, University of Tsukuba, Tsukuba, Japan. [10] These authors contributed equally: Haruki Nishimura, Mitsuhiro Yoshimura. [11] These authors jointly supervised this work: Mitsuhiro Yoshimura, Yoichi Ueta. ✉email: myoshim@med.uoeh-u.ac.jp; yoichi@med.uoeh-u.ac.jp

Oxytocin (OT), a neuropeptide synthesized in the hypothalamic supraoptic (SON) and paraventricular nuclei (PVN) and secreted from the posterior pituitary (PP) into the systemic circulation, elicits diverse actions in the peripheral and central nervous system (CNS). Different OT populations have been identified as magnocellular and parvocellular OT neurons. Magnocellular OT neurons in the SON and PVN (mPVN) affect adjacent neurons via somato-dendritic release and/or axonal projections to the PP[1]. On the other hand, parvocellular OT neurons, distributed throughout the dorsal PVN (dPVN), send projections to the CNS, notably including the spinal dorsal horn[2]. OT modulates social and sexual behaviors as well as sensory and autonomic functions through these neurosecretory systems, playing an important role in complex social interactions, such as maternal behavior, partnership, and social bonding[3]. Malfunction of the OT system is believed to be responsible for the impaired social behavior associated with autism, social anxiety, stress disorder, and schizophrenia[4].

Additionally, many ambitious studies have suggested a role for OT in pain modulation and anti-nociceptive effects. Analgesic effects of OT have also been hypothesized to be mediated by vasopressin V1a receptor (V1aR)[5,6]. Different OT-ergic pathways are assumed to be involved in its anti-nociceptive action; one is the descending pain inhibitory system in the CNS and the other is via an indirect effect on the dorsal root ganglia (DRG) in the periphery[7]. OT amplifies gamma amino butyric acid (GABA)-ergic inhibition, which hampers nociceptive signal transduction, at superficial and deep layers of spinal dorsal horn neurons[8]. OT also modulates hypothalamus–pituitary–adrenal (HPA) axis activity under stressful conditions[9]. In addition, OT is involved in the modulation of immune and inflammatory processes[10], which also might contribute to its analgesic effects. To date, we have demonstrated that OT is involved in the pain modulation using various experimental pain models in rats[11–13]. Elucidation of the direct effects of endogenous OT on pain pathways, however, has been particularly challenging due to technical limitations regarding the lack of specificity in OT neurons.

To overcome the aforementioned technical challenges, chemogenetics, also known as designer receptors exclusively activated by designer drugs (DREADDs) and optogenetics have been applied to OT neurons. Using these techniques, Eliava et al. demonstrated that selective parvocellular OT activation inhibited sensory processing via wide dynamic range (WDR) neurons, which resulted in analgesia in an inflammatory pain model in mice[14]. They also found that parvocellular OT neurons activated magnocellular OT neurons in the SON and increased OT release. Generally, viral transfection is used for exogenous gene expression, however, this method is fraught with technical challenges. In the present study, as an experimental refinement, we have generated a transgenic rat line that expresses human muscarinic acetylcholine receptor (hM3Dq, excitatory DREADDs) and mCherry exclusively in OT neurons.

A completely specific OT receptor (OTR) antagonist has proved elusive; therefore, it has been nearly impossible to confidently determine which receptor is responsible for the analgesic actions of OT. In this work, we have used the transgenic rats to provide compelling data to support the hypothesis that the analgesic effect of OT is mediated predominantly via OTR, although we still cannot completely rule out the possibility of a role for V1aR. Finally, using our OT-hM3Dq-mCherry transgenic rats, we provide direct evidence that endogenous OT is involved in anti-nociception and the anti-inflammatory response via the descending pain inhibitory system in the CNS and periphery.

## Results

### Generation of the OT-hM3Dq-mCherry transgenic rat line.
A chimeric OT-hM3Dq-mCherry bacterial artificial chromosome (BAC) clone transgene was constructed (Fig. 1a). hM3Dq-mCherry was exclusively expressed under the OT promoter. We observed mCherry fluorescence in the SON and PVN (Fig. 1b, c), site of OT synthesis. Therefore, ongoing expression of mCherry in these nuclei was used as a robust marker of transgene expression in the rat breeding line.

To confirm both mCherry and hM3Dq were expressed exclusively in OT-producing neurons, we performed fluorescent immunohistochemistry (FIHC) for vasopressin (VP), Fos, and OT in the SON and PVN at 120 min after the subcutaneous (s.c.) injection of CNO (1 mg kg$^{-1}$) (Fig. 1c–j). Confocal laser scanning microscopic observation revealed that the mCherry neurons were predominantly co-localized with OT-immunoreactive (-ir) neurons (SON, 98.2 ± 0.4%; PVN, 98.4 ± 0.4%) (Fig. 1d, f), whereas significantly fewer mCherry neurons co-localized with VP-ir neurons (SON, 2.1 ± 0.3%; PVN, 2.5 ± 0.3%) ($n = 12$ slices from 6 rats, each) (Fig. 1e, f). Indeed, 3–5% of magnocellular neurosecretory cells (MNCs) expressed equivalent levels of VP and OT[15], hence, our present results are consistent with the interpretation that mCherry is expressed in OT neurons, irrespective of whether the OT neurons also co-express VP. Fos expression, which is one of the markers for neuronal activation, was robustly increased in mCherry neurons (SON, 92.3 ± 1.3%; PVN, 91.6 ± 0.6%) (Fig. 1g, i), but not in VP neurons (SON, 3.0 ± 0.4%; PVN, 4.7 ± 0.7%) ($n = 12$ slices from 6 rats, each) (Fig. 1h, i). Three-dimensional reconstructed SON images consistently revealed that Fos-ir was exclusively expressed in OT-ir neurons (Fig. 1j). Taken together, these results suggested that successful establishment of the transgenic rat line had been achieved.

### OT-hM3Dq-mCherry was functioning in the transgenic rats.
FIHC for Fos was performed at 120 min after s.c. injection of Saline or CNO (1 mg kg$^{-1}$) ($n = 5$–6 rats, each), and the number of mCherry neurons co-expressed with Fos-ir (neurons with red cytoplasm and green nucleus) was manually counted. Robust Fos expression was observed in the SON (Fig. 2a, c) and PVN (Fig. 2b, c) after s.c. injection of CNO compared to Saline ($n = 10$–12 slices from 5 to 6 rats, each). For further investigation, PVN was anatomically separated into mPVN and dPVN (Fig. 2d). Fos induction was observed in both the mPVN and dPVN after s.c. CNO ($n = 10$–12 slices from 5 to 6 rats, each) (Fig. 2e). On the other hand, Fos-ir was not observed after s.c. injection of either Saline or CNO in wild type (WT) rats (Supplementary Fig. 1a–c), indicating that the effect was specific to CNO injection of OT-hM3Dq-mCherry transgenic rats. The transgene is not expressed in WT rats, and consistent with this, no mCherry fluorescence was detected in the SON and PVN of WT rats (Supplementary Fig. 1a, b). As expected, serum OT and VP both remained unchanged after s.c. CNO injection of WT rats (Supplementary Fig. 1d, e).

hM3Dq-mCherry was found to be localized in axons as well as in cell bodies of the OT neurons (Fig. 2f). In addition, mCherry was observed in the PP (Supplementary Fig. 2a). These results suggested that hM3Dq might be expressed in the axons and terminals as well as in the cell body of the OT neurons and may secrete OT via somato-dendritic release[1].

Both serum OT concentration and *oxytocin* (OXT) gene expression were assessed to investigate the dynamics of endogenous OT production after s.c. CNO injection. Serum OT concentration, analyzed by radioimmunoassay (RIA), was

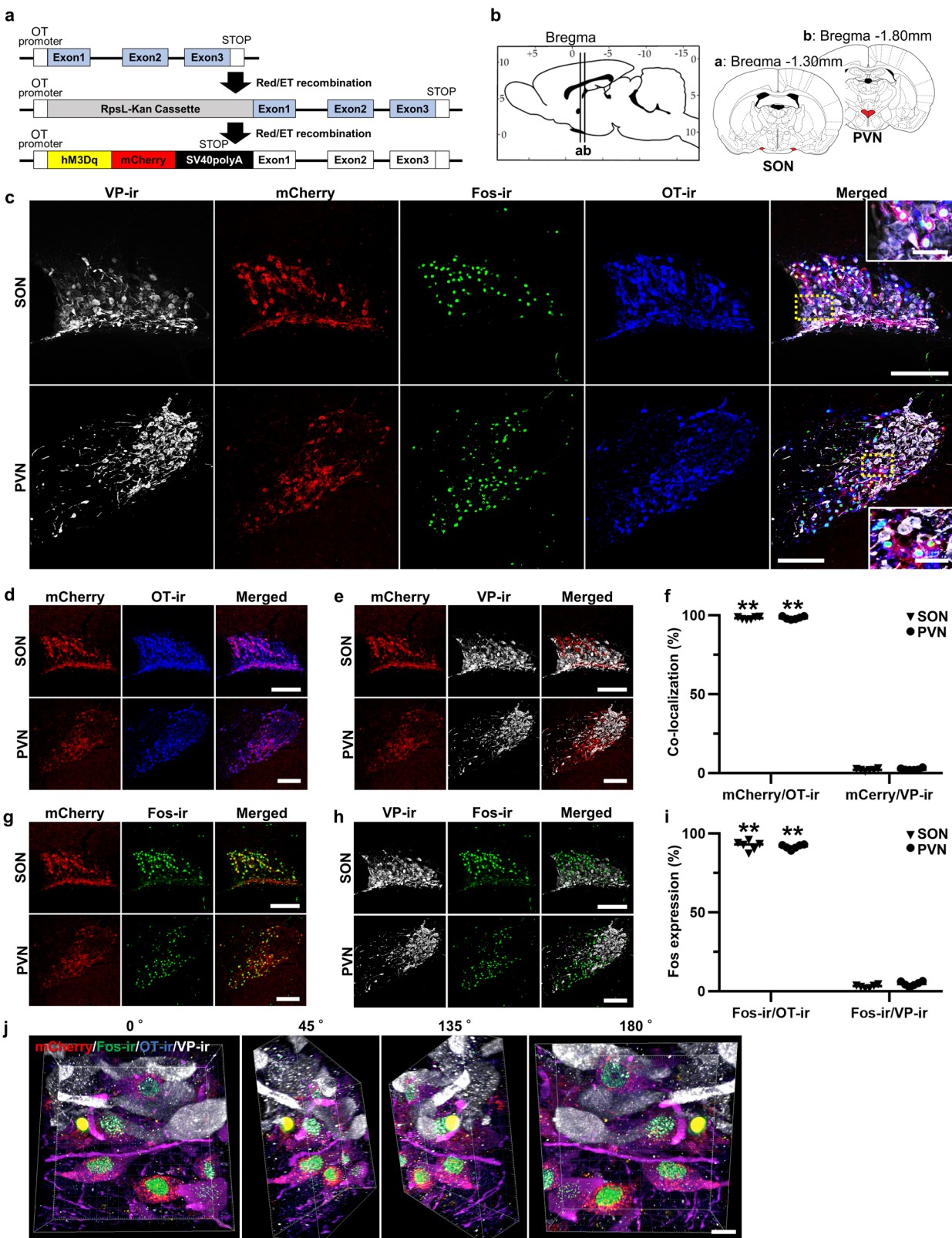

significantly elevated at 30, 60, 120, and 180 min after s.c. CNO injection ($1\,\text{mg}\,\text{kg}^{-1}$) in comparison to Saline ($n = 6\text{–}7$ rats in each group at each time point) (Fig. 2g). *OXT* gene expression in the SON, mPVN, and dPVN, analyzed by in situ hybridization histochemistry (ISH) (Fig. 2h), was also significantly increased at 120 and 180 min after s.c. CNO injection ($1\,\text{mg}\,\text{kg}^{-1}$) ($n = 12$ slices from 6 rats in each group at each time point)

(Fig. 2i). These results indicated that OT production, as well as OT secretion, was increased by chemogenetic activation of OT neurons.

On the other hand, s.c. CNO injection of OT-hM3Dq-mCherry transgenic rats did not affect gene expression of *arginine vasopressin* (*AVP*) in the hypothalamus nor did it affect serum VP concentration (Supplementary Fig. 2b–d), indicating that

**Fig. 1 Generation of the OT-hM3Dq-mCherry transgenic rat line. a** A concept of generating oxytocin (OT)-hM3Dq-mCherry transgenic rat. **b** Schematic illustration of the supraoptic (SON) and paraventricular nuclei (PVN). **c** Confocal images of vasopressin (VP)-immunoreactivity (ir), endogenous mCherry, Fos-ir, OT-ir, and merged images of the SON and PVN at 120 min after the subcutaneous (s.c.) injection of clozapine-N-oxide (CNO, 1 mg kg⁻¹). Rectangles framed by yellow dotted lines are enlarged in magnified images. **d** Confocal images of endogenous mCherry, OT-ir, and merged images of the SON and PVN. **e** Percentage of co-localization of endogenous mCherry, VP-ir, and merged images of the SON and PVN. **f** Co-localization of mCherry with OT-ir or VP-ir neurons in the SON and PVN ($n = 12$ slices from 6 rats, each). **g** Co-localization of endogenous mCherry, Fos-ir, and merged images of the SON and PVN at 120 min after the s.c. injection of CNO. **h** Co-localization of VP-ir, Fos-ir, and merged images of the SON and PVN at 120 min after the s.c. injection of CNO. **i** Percentage of Fos-ir positive neurons in mCherry positive neurons or VP-ir positive neurons at 120 min after the s.c. injection of CNO in the SON and PVN ($n = 12$ slices from 6 rats, each). **j** Three-Dimensional reconstructed image (endogenous mCherry (red), FIHC for Fos (green), OT (blue) and VP (white)) of the SON at 120 min after s.c. injection of CNO. Scale bars in (**a–h**), 200 µm and 50 µm (in magnified images). Scale bar in (**j**), 10 µm.

CNO did not appear to affect adjacent VP neurons in the transgenic rats.

**OT altered nociceptive behavior and activated noradrenergic neurons in the LC.** The von Frey and hot plate tests were carried out to assess the effect of endogenous OT on mechanical/heat sensitivities in naïve transgenic rats. Both tests were carried out at 0, 30, 60, 120 and 180 min after the s.c. injection of Saline or CNO (1 mg kg⁻¹) ($n = 11$ rats, each). Withdrawal threshold, tested by von Frey filament, was significantly elevated at 30 min after s.c. injection of CNO compared to Saline (Fig. 3a). The latency of nocifensive behavior, evaluated by hot plate test, was markedly prolonged in rats after s.c. CNO injection compared to s.c. Saline-injected controls (Fig. 3b). Mechanical/heat sensitivities were not altered in WT rats after the s.c. injection of CNO, indicating that CNO alone did not affect nociceptive behaviors (Supplementary Fig. 4a, b).

We hypothesized that the anti-nociceptive effects of endogenous OT did arise from the activation of the descending pain inhibitory system. Fos expression was analyzed in the locus coeruleus (LC) (Fig. 3c) and dorsal raphe nucleus (DR) (Fig. 4a), two nuclei that are involved in the descending pain inhibitory system[16]. Tyrosine hydroxylase (TH) and tryptophan hydroxylase (TPH) are the rate-limiting enzyme in the biosynthesis of noradrenaline and serotonin (5-hydroxytryptamine; 5-HT), respectively[17,18]. TH is expressed in noradrenergic neurons in the LC, and TPH2 is expressed in serotonergic neurons localized within the dorsal and median raphe nuclei of the CNS. We analyzed TH-ir and TPH-ir neurons to evaluate the expression of noradrenergic and serotonergic neurons in the LC (Fig. 3d) and DR (Fig. 4b), respectively. The number of TH-ir neurons was comparable between Saline and CNO (1 mg kg⁻¹), whereas the percentage of TH-ir neurons co-expressed with Fos-ir was significantly increased after s.c. CNO injection (1 mg kg⁻¹) ($n = 10–12$ slices from 5 to 6 rats, each) (Fig. 3f). ISH analysis of *TH* (Fig. 3e) revealed that the gene expression of *TH* was significantly increased at 120 and 180 min after s.c. injection of CNO (1 mg kg⁻¹) compared to Saline ($n = 12$ slices from 6 rats, each) (Fig. 3g). Typical representative FIHC images of the LC after s.c. injection with Saline or CNO in the transgenic rats in transgenic rats are shown (Supplementary Fig. 3a).

**OT-activated serotonergic neurons in the DR and inhibitory interneurons in the spinal dorsal horn.** The DR was divided into four following subregions; ventral (DRv), inter-fascicular (DRi), dorsal (DRd) and ventrolateral wings (DRvl) since each sub-nucleus exhibits differential neuroanatomical and distinct functional roles[19] (Fig. 4b). The number of TPH-ir neurons, Fos-ir neurons, and TPH-ir neurons co-expressed with Fos-ir neurons were significantly increased after s.c. CNO injection (1 mg kg⁻¹) ($n = 10–12$ slices from 5 to 6 rats, each) (Fig. 4d). The gene expression of *TPH2* (Fig. 4c) in each subregion was also significantly increased ($n = 12$ slices from 6 rats, each) (Fig. 4e), a

finding compatible with our FIHC results. Typical representative FIHC images of the DR after treatment with Saline or CNO in the transgenic rats are shown (Supplementary Fig. 3b).

FIHC for Fos and PAX2, which is a transcription factors that is exclusively expressed in the inhibitory interneurons in the spinal cord[20], was carried out in the dorsal horn of lumbar segment 5 (L5) (Fig. 4f) in the spinal cord (Fig. 4g). The dorsal horn was divided into the superficial layer (laminae I–II) and deep layer (laminae III–VI), as differential roles have been proposed depending on the region[21]. The number of PAX2-ir neurons, Fos-ir neurons, and co-expressed PAX2-ir/Fos-ir neurons in the superficial layer and deep layer were dramatically increased in CNO-injected transgenic rats (1 mg kg⁻¹) compared to Saline-injected transgenic rats ($n = 10–12$ slices from 5 to 6 rats, each) (Fig. 4h, i).

In the LC, DR, and spinal dorsal horn, most regions were not affected by the s.c. injection of CNO (1 mg kg⁻¹) in WT rats (Supplementary Fig. 4c–j). The number of TPH-ir neurons, however, was statistically increased after the s.c. injection of CNO in the DRv and DRvl of WT rats without altering the number of Fos-ir neurons (Supplementary Fig. 4g). This may be one of the off-target effects of CNO[22]. CNO contains a similar structure of clozapine, an antipsychotic drug that may lead to an up-regulation of TPH[23]. Although the concentration of CNO used in our study is widely used for chemogenetic activation, and should not have major off-target effects, it is possible that it could induce the small but significant increase in TPH-ir neurons we observed in WT rats after CNO injection. In addition, several kinds of physical stress can increase the expression of TPH[24]. Of note, however, we did not see any significant increase of TPH-ir + Fos-ir neurons in the DR of WT rats (Supplementary Fig. 4g).

**OT induced differential gene expression in GABA-ergic neurons of the dorsal horn.** Using retrograde tracer, we have confirmed direct and indirect neuronal projections into the L5 dorsal horn from the dPVN, LC and DR (Supplementary Fig. 5a–f). In addition, using our OT-monomeric red fluorescent protein 1 (mRFP1) transgenic rats[25], axons of OT neurons were detected in the LC, DR and the dorsal horn of the L5 spinal cord (Supplementary Fig. 5g). The results indicate that OT-ergic projections might directly and indirectly modulate pain transmission in the spinal dorsal horn.

We speculated that functional gene expression pathways related to anti-nociception might be changed after OT activation, since abundant innervations from OT^PVN neurons were observed in the spinal dorsal horn. We therefore employed RNAseq to investigate this hypothesis in an unbiased manner, rather than using a targeted candidate approach. The dorsal horn, divided into a superficial layer (laminae I–II) and deep layer (laminae III–VI), was micro-dissected and analyzed separately ($n = 30$ specimens from 3 rats, each) (Supplementary Fig. 6a). Consistent RNAseq mapping patterns throughout the full length

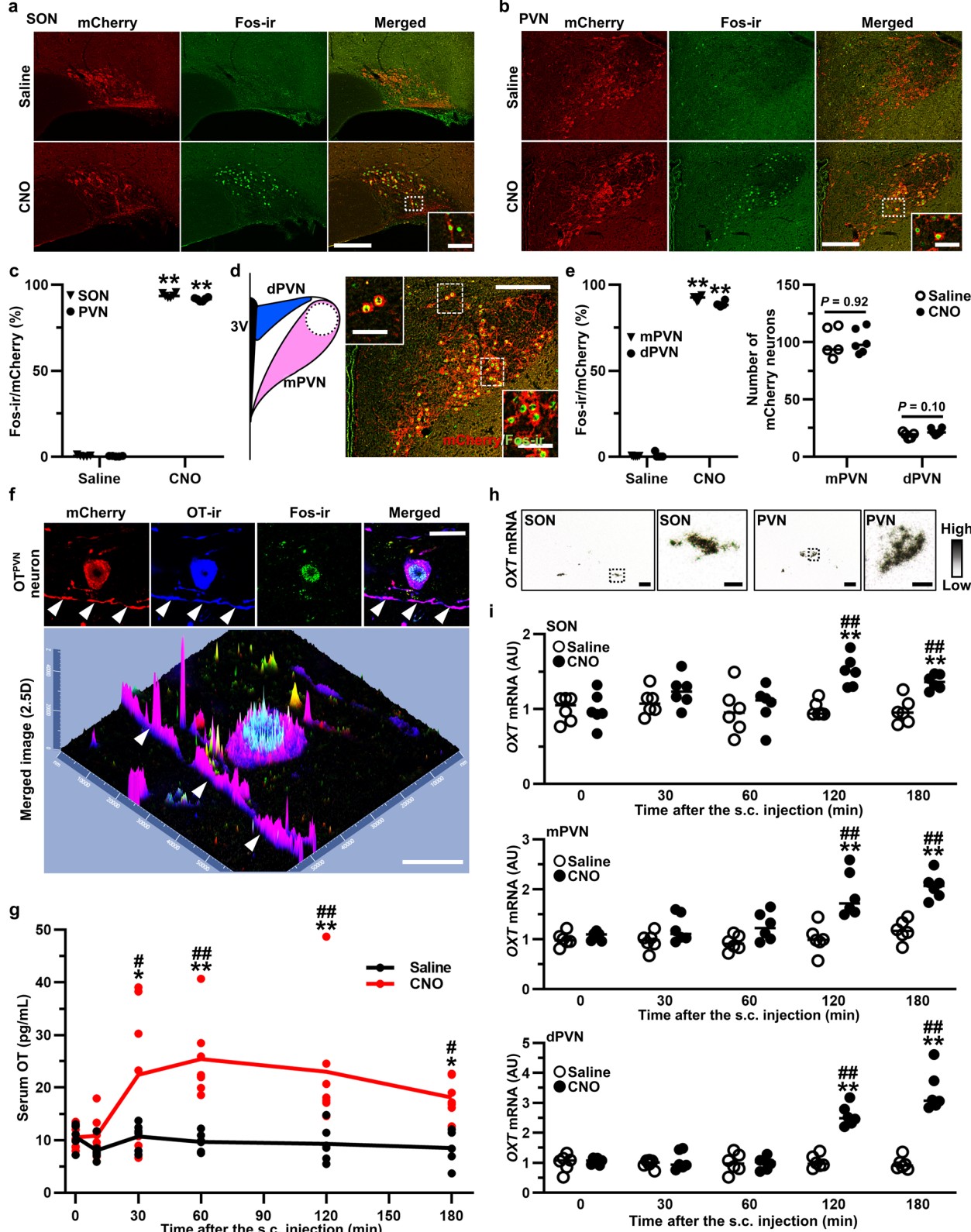

of known housekeeping genes, along with consistent read counts across the different samples indicated good data quality (Supplementary Fig. 6b–d).

In the laminae I–II, 254 genes were significantly differentially expressed ($p$ value < 0.05, |fold change| > 2) after chemogenetic activation of endogenous OT. On the other hand, 191 genes were significantly differentially expressed ($p$ value < 0.05, |fold change| > 2) in the laminae III–VI (Fig. 5a–d). Interestingly, many of these genes have previously been shown to be predominantly expressed in GABA-ergic neurons[26]. It is suggested that endogenous OT might directly and/or indirectly inhibit local afferent sensory transmission. Gene ontology (GO) enrichment analysis was performed for interpreting the functional categories of these altered genes in the dorsal horn (Fig. 5e,

**Fig. 2 OT-hM3Dq-mCherry was functioning in the transgenic rats.** Endogenous mCherry, Fos-ir and merged images of the SON (**a**) and PVN (**b**) at 120 min after the s.c. injection of Saline or CNO (1 mg kg$^{-1}$). **c** Percentage of Fos-ir neurons in mCherry neurons in the SON and PVN (n = 10–12 slices from 5 to 6 rats, each). **P < 0.01 vs. Saline. **d** The PVN was anatomically divided into magnocellular PVN (mPVN) and dorsal parvocellular PVN (dPVN). **e** Percentage of Fos-ir neurons in the mPVN and dPVN (n = 10–12 slices from 5 to 6 rats, each). **P < 0.01 vs. Saline. **f** Confocal images of endogenous mCherry (red), OT-ir (blue), Fos-ir (green) and merged images of a single OT$^{PVN}$ neuron at 120 min after the s.c. injection of CNO. White arrow heads, dendrite of OT$^{PVN}$ neuron. **g** The serum OT concentration at 0, 10, 30, 60, 120, and 180 min after the s.c. injection of Saline or CNO (n = 6–7 rats in each group at each time point). *P < 0.05; **P < 0.01 vs. Saline at the same time point. #P < 0.05; ##P < 0.01 vs. CNO at 0 min. **h** In situ hybridization (ISH) histochemistry of *oxytocin* (*OXT*) in the SON and PVN. **i** Gene expression of *OXT* in the SON, mPVN, and dPVN at 0, 30, 60, 120, and 180 min after the s.c. injection of Saline or CNO (n = 12 slices from 6 rats, each). **P < 0.01 vs. Saline at the same time point. ##P < 0.01 vs. CNO at 0 min. Scale bars in (**a**, **b**, **d**, **h**, 200 μm and 50 μm (in magnified images). Scale bars in (**f**), 10 μm. See also Supplementary Figs. 1, 2.

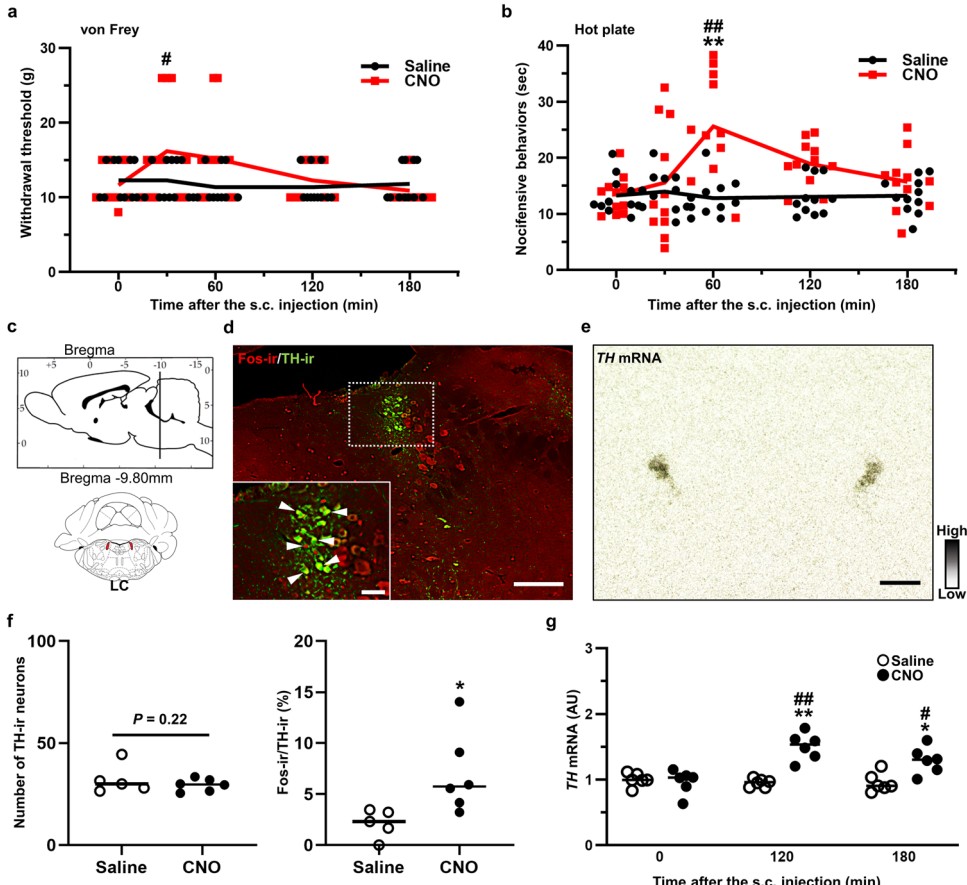

**Fig. 3 OT altered nociceptive behavior and activated noradrenergic neurons in the LC.** Results of manual von Frey (**a**) and hot plate tests (**b**). **c** Schematic illustration of the locus coeruleus (LC). **d** FIHC images of tyrosine hydroxylase (TH)-ir and Fos-ir in the LC at 120 min after the s.c. injection of CNO. **e** ISH image of *TH* in the LC. **f** Number of TH-ir neurons and percentage of Fos-ir neurons in TH-ir neurons in the LC (n = 10–12 slices from 5 to 6 rats, each). **g** Gene expression of *TH* in the LC after the s.c. injection of Saline or CNO (1 mg kg$^{-1}$) (n = 12 slices from 6 rats, each). Scale bars, 200 μm. Data are represented as mean ± SEM. *P < 0.05; **P < 0.01 vs. Saline. #P < 0.01; ##P < 0.01 vs. CNO at 0 min. See also Supplementary Figs. 3, 4.

f). GO terms that were related to "inflammatory response" or "immune response" were enriched after the chemogenetic activation of OT in the spinal dorsal horn. It therefore now appears that endogenous OT might also modulate the spinal inflammation and/or immune response, resulting in suppressing sensory transmission. Raw RNA sequencing data are available at Gene Expression Omnibus (GEO) in the National Center for Biotechnology Information (NCBI) (accession number GSE210528).

**OT alleviated hyperalgesia in a neuropathic pain model via neuronal pathway.** We employed the Seltzer model which is developed by partial ligation of the right sciatic nerve as a neuropathic pain model[27] (Fig. 6a). This model enables us to evaluate

the effect of OT on neuropathic pain since hypersensitivities against mechanical and heat stimulation develop rapidly, and continue for a substantial amount of time. Withdrawal threshold (mechanical sensitivity) and the latency of nocifensive behavior (heat sensitivity to 52.5 °C) were promptly decreased after the Seltzer surgery (Supplementary Fig. 7a, b). In addition, we have confirmed the effects of CNO (1 mg kg$^{-1}$) in Sham-operated animals at post-operative day 10. The results were consistent with naïve transgenic rats (Supplementary Fig. 7c, d). To exclude unwanted effects of surgical intervention, the experiment was performed at 10 days after the surgery. Mechanical/heat sensitivities were evaluated for 3 h after administering Saline or CNO (1 mg kg$^{-1}$) (Fig. 6b). Withdrawal threshold was significantly elevated at 0.5, 1, and 2 h after the s.c. injection of CNO (Fig. 6d),

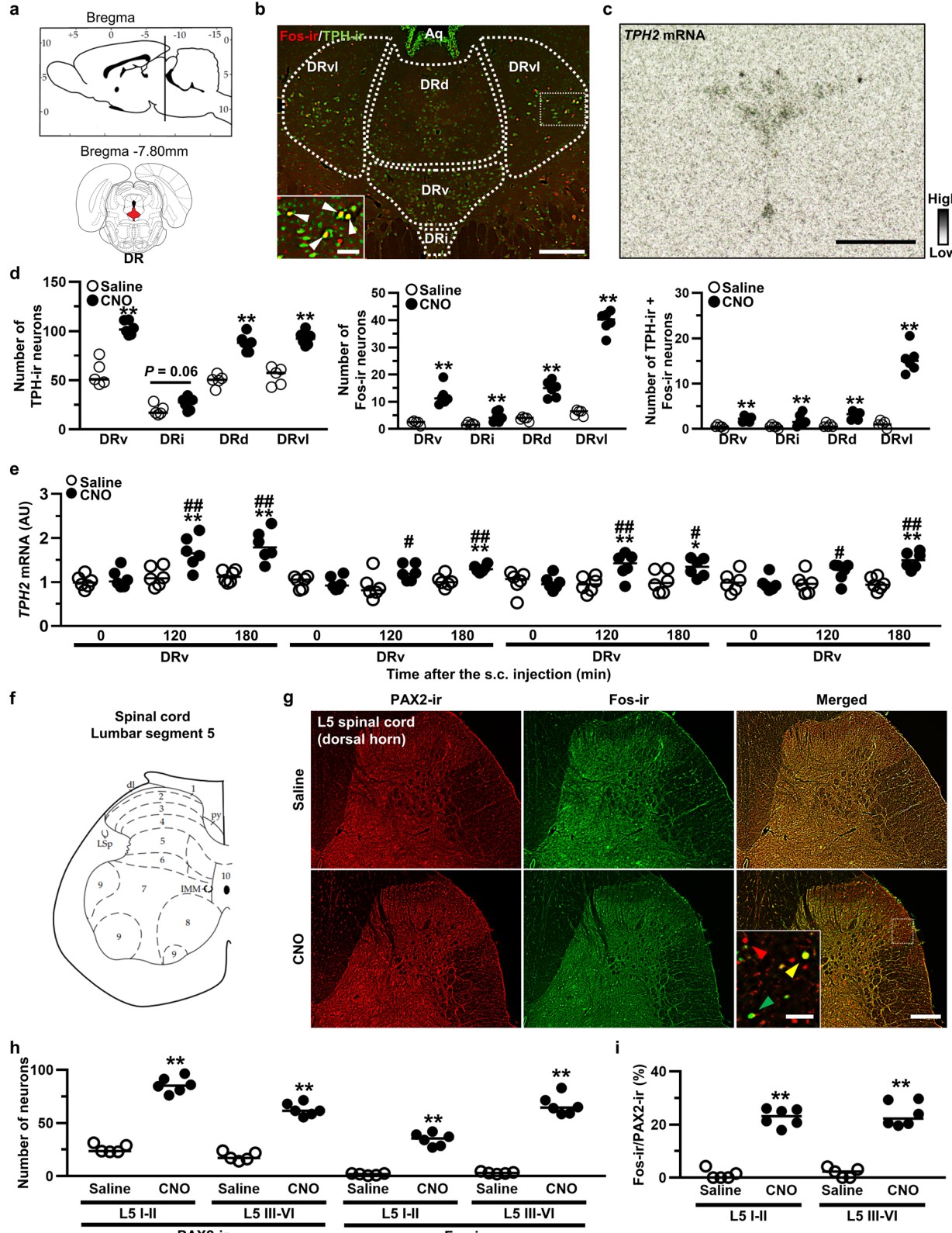

whilst the latency of nocifensive behavior was markedly prolonged at 1 h after the s.c. injection of CNO compared to Saline ($n = 9$ rats, each) (Fig. 6e).

Next, we used OTR antagonist to examine whether the antihyperalgesic effects of OT on neuropathic pain were ascribed to neuronal or humoral targets of OT (Fig. 6c). Rats were either treated with an intraperitoneal (i.p.) injection of OTR antagonist

(L-371,257, dissolved in dimethyl sulfoxide (DMSO) [$10 \, mg \, kg^{-1}$]), intrathecal (i.t.) injection of OTR antagonist (Atosiban, dissolved in saline [$1 \, \mu g \, \mu L^{-1}$, $12 \, \mu g$ per rat]) or both ($n = 9$ rats, each). Since OT itself does not cross the blood brain barrier (BBB), the role of neuronal/humoral OT could be differentiated by using these antagonists. The dose of i.t. OTR antagonist was very small that the effects on periphery was negligible. Thereafter, all rats were

**Fig. 4 OT-activated serotonergic neurons in the DR and inhibitory interneurons in the spinal dorsal horn. a** Schematic illustration of the dorsal raphe nucleus (DR). **b** Tryptophan hydroxylase (TPH)-ir and Fos-ir in the DR at 120 min after the s.c. injection of CNO (1 mg kg$^{-1}$). DR were divided into ventral (DRv), inter-fascicular (DRi), dorsal (DRd) and ventrolateral "wings" (DRvl). **c** ISH image of *TPH2* in the DR. **d** TH-ir neurons, Fos-ir neurons and percentage of their co-expression in the DR ($n = 10$–12 slices from 5 to 6 rats, each). **e** Gene expression of *TPH2* in the DR after the s.c. injection of Saline or CNO ($n = 12$ slices from 6 rats, each). **f** Schematic illustration of the lumbar segment 5 (L5) in the spinal cord. **g** PAX2-ir, Fos-ir and merged images in the L5 spinal cord at 120 min after the s.c. injection of CNO. **h** PAX2-ir neurons in the superficial layer (laminae I–II) and deeper layer (laminae III–VI) of the L5 ($n = 10$–12 slices from 5 to 6 rats, each). **i** Percentage of co-localization of Fos-ir and PAX2-ir neurons in the L5 ($n = 10$–12 slices from 5 to 6 rats, each). Scale bars, 200 μm and 50 μm (in magnified images). *$P < 0.05$; **$P < 0.01$ vs. Saline. #$P < 0.01$; ##$P < 0.01$ vs. CNO at 0 min. See also Supplementary Figs. 3, 4.

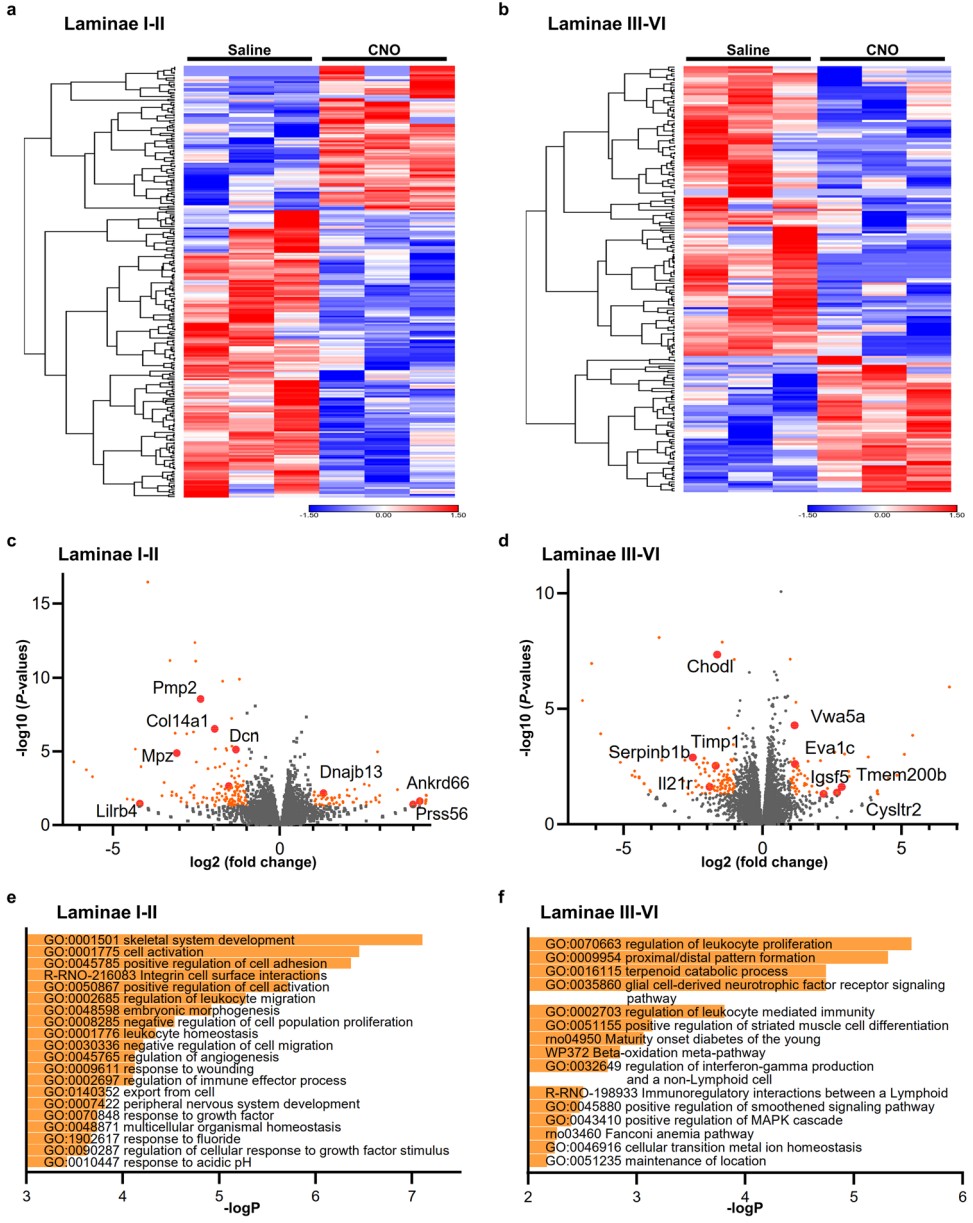

**Fig. 5 OT induced differential gene expression in GABA-ergic neurons of the dorsal horn.** Heat map of the altered genes in the laminae I–II (**a**) and laminae III–VI (**b**) of the spinal cord after the chemogenetic activation of endogenous OT by using RNAseq. The samples for the analysis were collected at 120 min after the s.c. injection of Saline or CNO (1 mg kg$^{-1}$). Scale represents normalized values subtracted by row mean divided by standard deviation. Volcano plot of the genes in the laminae I–II (**c**) and laminae III–VI (**d**) of the spinal cord. The genes that were significantly altered were depicted by orange color dots. Red-colored dots indicate subset of the genes that are predominantly expressed in GABA-ergic neurons. Gene ontology (GO) terms that were enriched by endogenous OT in the laminae I–II (**e**) and laminae III–VI (**f**) of the spinal cord. Raw RNA sequencing data are available at Gene Expression Omnibus (GEO) in the National Center for Biotechnology Information (NCBI) (accession number GSE210528). See also Supplementary Figs. 5, 6.

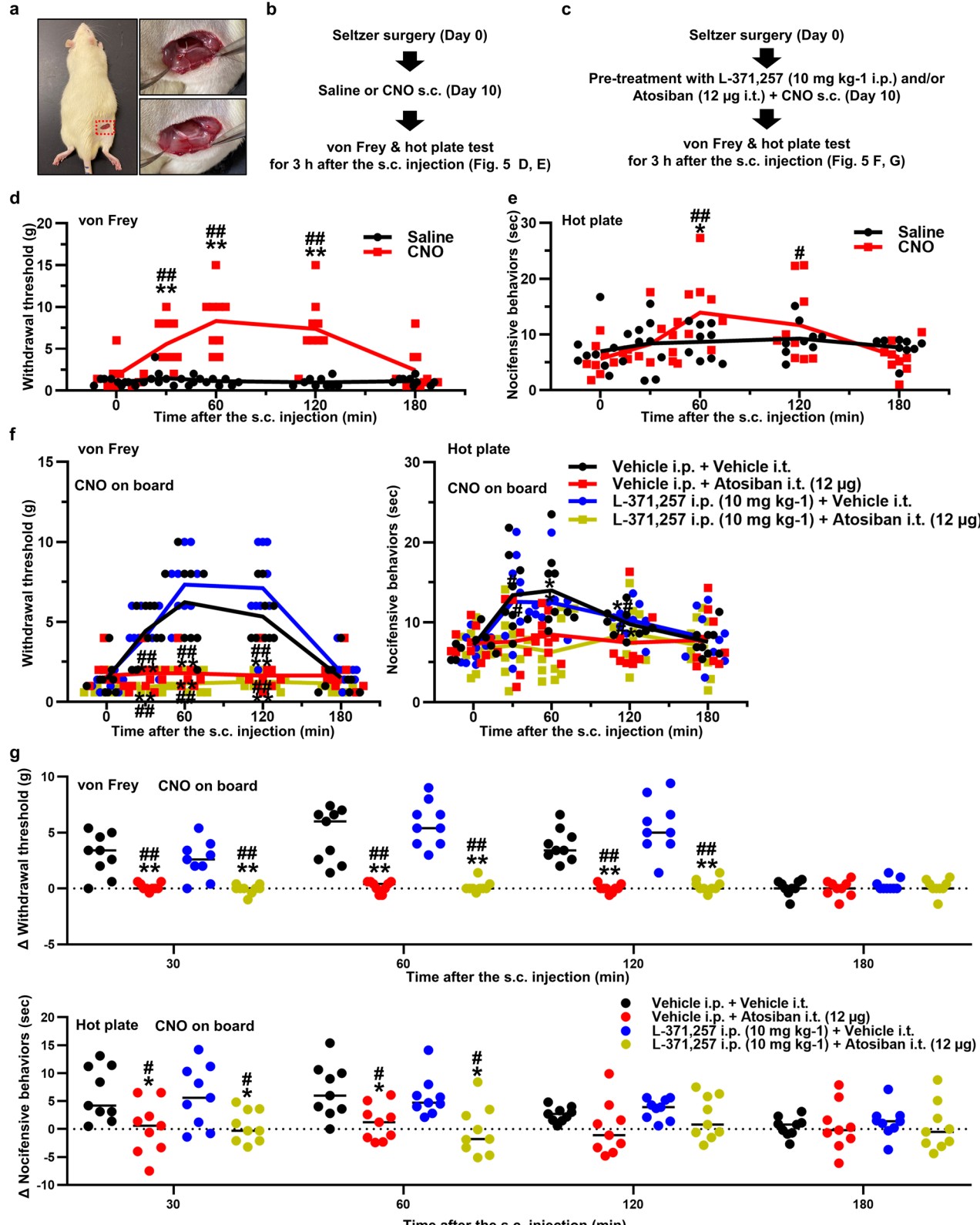

**Fig. 6 OT alleviated hyperalgesia in a neuropathic pain model via neuronal pathway. a** The Seltzer surgery was performed via dorsal approach. Experimental procedures administering CNO after the Seltzer surgery (**b**) and OT receptor (OTR) antagonist treatment (**c**). Results of the manual von Frey (**d**) and hot plate tests (**e**) after the s.c. injection of Saline or CNO ($1\,mg\,kg^{-1}$) in the Seltzer model ($n = 9$ rats, each). Data are represented as mean ± SEM. **$P < 0.01$ vs. Saline at the same time point. #$P < 0.05$; ##$P < 0.01$ vs. CNO at 0 min. Results of the manual von Frey (**f**) and hot plate tests (**g**) after the s.c. injection of CNO ($1\,mg\,kg^{-1}$) pretreatment with L-371,257 dissolved in DMSO [$10\,mg\,kg^{-1}$] for OTR antagonist i.p., Atosiban dissolved in saline [$1\,\mu g\,\mu L^{-1}$] for OTR antagonist i.t., and/or vehicle in the Seltzer model ($n = 9$ rats, each). Data are represented as mean ± SEM. **$P < 0.01$ vs. Vehicle i.p. + Vehicle i.t., #$P < 0.05$; ##$P < 0.01$ vs. OTR antagonist i.p. + Vehicle i.t. at the same time point. See also Supplementary Fig. 7.

subcutaneously injected with CNO (1 mg kg$^{-1}$). Mechanical/heat sensitivities were evaluated for 3 h. Vehicle i.p. and i.t. treatment did not affect both withdrawal threshold and the latency of nocifensive behavior, whilst combined i.p. plus i.t. injection of OTR antagonist ablated the effects of CNO (Fig. 6f, g), suggesting that endogenous OT was involved in the alteration of mechanical/heat sensitivities. Interestingly, the effects of CNO on withdrawal threshold and the latency of nocifensive behavior were not affected by i.p. injection of OTR antagonist, but was completely abolished by i.t. injection of OTR antagonist (Fig. 6f, g). These results suggested that the neuronal OT-ergic pathway plays a greater role in the inhibition of mechanical/heat transmissions, than the humoral OT pathway does, at least in the neuropathic pain model.

**OT alleviated spontaneous nociceptive behaviors and hyperalgesia in inflammatory pain models via the humoral pathway.** We also examined the effect of endogenous OT on the spontaneous nociceptive behaviors using the formalin test model[28] (Fig. 7a). Strikingly, the right foot pad swelling that resulted from s.c. injection of 5% formalin (100 µL) was significantly attenuated by pretreatment with CNO (1 mg kg$^{-1}$) ($n = 11$ rats, each) (Fig. 7c, d). Total licking time (analyzed at 5 min intervals for 60 min) was significantly decreased in CNO compared to Saline ($n = 5$–6 rats, each) (Fig. 7e). This decrease was detected in both the 1st and 2nd phase (Fig. 7f).

As with the Seltzer model, OTR antagonist was administered (Fig. 7b). Rats were either treated with i.p. injection of OTR antagonist (L-371,257 [10 mg kg$^{-1}$]), i.t. injection of OTR antagonist (Atosiban [1 µg µL$^{-1}$, 12 µg per rat]) or both ($n = 6$ rats, each). Thereafter, all rats were subcutaneously injected with CNO (1 mg kg$^{-1}$). Total licking time (analyzed at 5 min intervals for 60 min) was then measured. As observed in the neuropathic pain model, vehicle i.p. plus i.t. treatment did not affect licking time, whereas combined i.p. plus i.t. injection of OTR antagonist ablated the effect of CNO (Fig. 7g). Surprisingly, the effects of CNO on licking time were not affected by i.t. injection of OTR antagonist, but was ablated by i.p. injection of OTR antagonist (Fig. 7g, h). These findings were in contrast to the results observed in the neuropathic pain model. Furthermore, the effects of CNO on right foot pad swelling was also abolished by the i.p. injection of OTR antagonist (Fig. 7i). The results demonstrated that humoral OT might play a greater role in the reduction of spontaneous nociceptive behaviors and inflammations.

We also tested the effects of endogenous OT on carrageenan knee arthritis model which is one of the classical inflammatory pain models[29]. Knee inflammation, along with mechanical/heat hypersensitivities, developed rapidly after the intraarticular (i.a.) injection of carrageenan (0.1 mL of 3% λ-carrageenan) and lasted for at least 24 h (Supplementary Fig. 8a–c). After the development of knee arthritis at 3 h after the i.a. injection of carrageenan, either Saline or CNO (1 mg kg$^{-1}$) was subcutaneously injected, then, mechanical/heat sensitivities and knee diameter were measured for 24 h. As with the formalin model, significant attenuation of mechanical/heat hypersensitivities and alleviation of knee swelling were observed after CNO treatment (Supplementary Fig. 8d–g).

**OT exerted an anti-inflammatory response by inhibiting mast cell degranulation.** The results from the inflammatory pain model experiment led us to speculate that, peripheral OT, rather than central OT, may be more relevant to the anti-inflammatory response, especially for the reduction of local swelling. Peripheral OT may exert an anti-nociceptive response by inhibiting local inflammatory responses both directly and indirectly. Histological analysis of right foot pad revealed that the thickness of the

subcutaneous tissue was significantly attenuated by pretreatment with CNO (1 mg kg$^{-1}$) ($n = 3$–5 rats, each) (Fig. 8a, c). In toluidine blue staining, the morphology of the granules in the mast cells appeared distinctly different in the CNO-pretreated group compared to the Saline-pretreated group. Therefore, chemogenetic activation of OT appeared to inhibit degranulation from the subcutaneous mast cells ($n = 5$ rats, each) (Fig. 8b, d), potentially explaining why peripheral OT suppressed local inflammation. Using a transmission electron microscope (TEM), abundant granules were seen in the mast cells of CNO-pretreated group, whilst more degranulated mast cells were observed in Saline-pretreated group after the injection of formalin (Fig. 8e).

For further confirmation, a mast cell stabilizer (Disodium cromoglicate (DSCG) dissolved in saline [50 mg mL$^{-1}$]) was used and compared with CNO (1 mg kg$^{-1}$). At 30 min prior to the test, Saline, CNO (1 mg kg$^{-1}$), or DSCG (50 mg kg$^{-1}$) was s.c. injected, then 5% formalin (100 µL) was injected into the right hind paw. Strikingly, the effects of DSCG on the formalin test were comparable to CNO (Supplementary Fig. 9a–c). These results might strengthen the mast cell hypothesis.

**OT affected hypothalamus–pituitary–adrenal (HPA) axis under pathological condition.** Since significant attenuation of inflammation was observed after pretreatment with CNO in the formalin test, we speculated that the HPA axis, which plays an important role in the anti-inflammatory response, might be modified by endogenous OT. However, no alteration was detected for either gene expression of *corticotropin-releasing hormone* (*CRH*) or *pro-opiomelanocortin* (*POMC*) after s.c. CNO injection of transgenic rats ($n = 12$ slices from 6 rats, each) (Fig. 9a–c). We further confirmed a lack of increase in Fos-ir in CRH-ir neurons after the chemogenetic activation of OT ($n = 12$ slices from 6 rats, each) (Fig. 9d, e). In addition, the serum concentration of adrenocorticotropic hormone (ACTH) (Fig. 9f) and corticosterone (CORT) (Fig. 9g) remained unchanged after chemogenetic activation of OT in naïve transgenic rat ($n = 6$ rats in each group at each time point). Therefore, it seems that the HPA axis may not play a role in the anti-inflammatory response induced by OT under naïve condition.

We also measured serum OT, CRH, adrenocorticotropic hormone (ACTH), and CORT under the pathological condition, as we speculated that OT might exert greater effects on the HPA axis in the stressful context of physical trauma and/or injury. Immediately after the 5% formalin injection (100 µL) into the right hind paw, Saline or CNO (1 mg kg$^{-1}$) was s.c. injected, then blood concentrations were measured by RIA or ELISA ($n = 5$ rats in each group at each time point). OT elevation after the s.c. injection of CNO (1 mg kg$^{-1}$) was confirmed (Fig. 9h). Although serum CRH and ACTH were not altered, CORT was significantly increased at 1 h after the s.c. injection of CNO (1 mg kg$^{-1}$) (Fig. 9i–k). In addition, restraint stress was performed to explore the modulative effects of OT on HPA axis in different transgenic rats. Although restraint stress itself caused significant elevation of OT, an additional effect of CNO on serum OT was also observed (Supplementary Fig. 10). However, no alteration of HPA axis was observed after acute restraint stress (Supplementary Fig. 10), indicating that endogenous OT may play differential roles in modulating the HPA axis, depending on the type or duration of stress.

A model describing the putative mechanism of endogenous OT on pain modulation, supported by our finding, is illustrated schematically (Fig. 10).

## Discussion

We have generated a transgenic rat line that expresses hM3Dq and mCherry specifically in OT neurons in the SON and PVN.

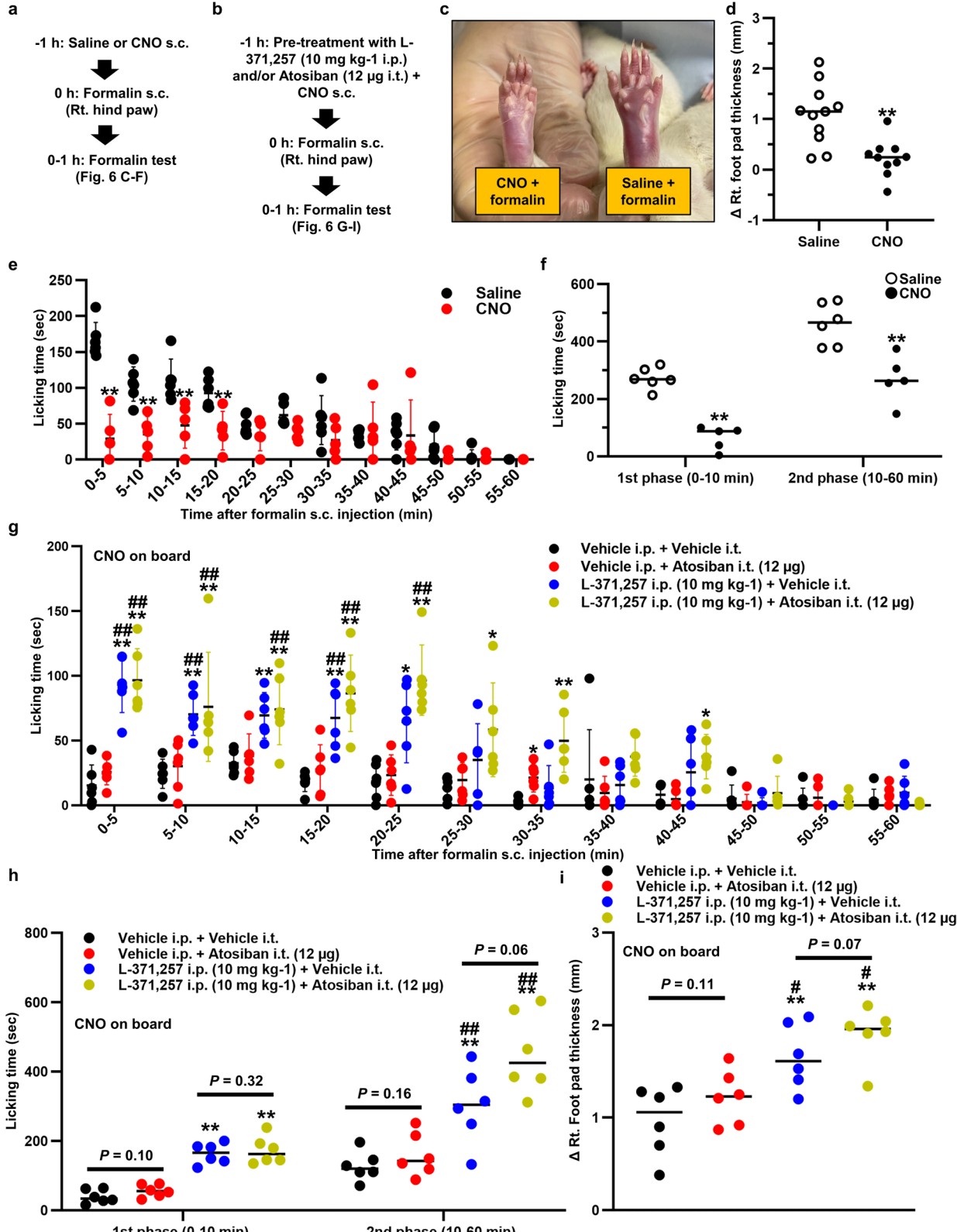

Endogenous OT was involved in pain transmission via both neuronal and humoral pathways. OT might exert anti-nociceptive, anti-hyperalgesia, and anti-inflammatory effects, indicating that this fascinating nonapeptide could be one of the potential therapeutic candidates for various pain- and inflammatory-related diseases.

We primarily focused on the LC, DR, and spinal dorsal horn as they play the crucial roles in pain transmission, although OT also acts on other parts of brain such as the periaqueductal gray, rostral ventromedial medulla, parabrachial nucleus, and amygdala. OT neurons activate GABA-ergic interneurons and may suppress pain signals. Indeed, OT receptors are expressed in the LC and DR[30]. Using optogenetic technique, it has been shown that OT is released into the LC from hypothalamic PVN OT fibers, activating noradrenergic neurons by co-release of OT and

**Fig. 7 OT alleviated spontaneous nociceptive behaviors and hyperalgesia in inflammatory pain models via the humoral pathway.** Experimental procedures administering CNO after the formalin test (**a**) and OT receptor (OTR) antagonist treatment (**b**). **c** Hind paw either pretreated with Saline or CNO. **d** Pretreatment with CNO significantly attenuated the swelling caused by 5% formalin (100 μL) injection ($n = 11$ rats, each). **e** Licking time was analyzed every 5 min interval for 60 min after the s.c. injection of 5% formalin (100 μL) ($n = 5$–6 rats, each). **P < 0.01 vs. Saline. **f** Total licking time was divided into 1st and 2nd phase ($n = 5$–6 rats, each). **P < 0.01 vs. Saline. **g** Licking time was analyzed at 5 min interval for 60 min after the s.c. injection of CNO (1 mg kg$^{-1}$) pretreatment with L-371,257 dissolved in DMSO [10 mg kg$^{-1}$] for OTR antagonist i.p., Atosiban dissolved in saline [1 μg μL$^{-1}$] for OTR antagonist i.t., and/or vehicle in the formalin test ($n = 6$ rats, each). Data are represented as mean ± SEM. **P < 0.01 vs. Vehicle i.p. + Vehicle i.t., #P < 0.05; ##P < 0.01 vs. Vehicle i.p. + OTR antagonist i.t. at the same time point. **h** Total licking time was divided into 1st and 2nd phase ($n = 6$ rats, each). **P < 0.01 vs. Vehicle i.p. + Vehicle i.t. **i** Foot pad thickness was measured at the start and the end of the formalin test and the difference was calculated ($n = 6$ rats, each). Data are represented as mean ± SEM. **P < 0.01 vs. Vehicle i.p. + Vehicle i.t., #P < 0.05, vs. Vehicle i.p. + OTR antagonist i.t. See also Supplementary Fig. 8.

glutamate[31]. Noradrenergic neurons in the LC and serotonergic neurons in the DR are important transitional nuclei for the descending pain inhibitory system[32,33]. The administration of local anesthetic directly into the LC contributed to alleviate neuropathic pain[32]. Abundant serotonergic neurons and other neuro-transmitters and/or neuro-modulators containing neurons are expressed in the DR. Their direct, or indirect via the nucleus raphe magnus, descending projections modulate the responses caused by noxious stimulation of the spinal dorsal horn neurons. On the other hand, their ascending projections directly modulate the responses of pain sensitive neurons in the thalamus. Arcuate nucleus of the hypothalamus may also be involved in analgesic effects[34]. Interestingly, Fos expression in TPH positive neurons was the most significantly increased in the lateral wing of the DR which is a stress-sensitive region. Serotonergic neurons in this area contribute to adoptive response to stress[35]. Also, TPH positive neurons located in the lateral wing area participate in modulating pain signals[19]. It is thus likely that endogenous OT may stimulate these neurons, resulting in an altered nociceptive threshold. In the present study, CNO induced a transient analgesia in naïve and sham Seltzer model rats; a different finding to the observation by Eliava et al.[14] and Iwasaki et al. (bioRxiv2022, https://doi.org/10.1101/2022.02.23.481531). Presumably, the different outcome between these studies is methodological, with our approach targeting a larger population of OT neurons rather than selectively targeting the neuronal circuit to the spinal cord.

The neuronal networks from dPVN, LC and DR to L5 spinal dorsal horn was confirmed using a retrograde tracer, consistent with results from previous studies[36,37]. OT innervation was more prominent between L4 and L6 in the superficial layers (laminae I–II). Given that OT is possibly involved in modulating pain processing, the neuronal networks is convincing.

Of note, many genes that were expressed in GABA-ergic neurons were altered by the chemogenetic activation of OT. Breton et al. however, have reported that OT-activated spinal cord neurons were exclusively non-GABA-ergic neurons in lamina II of acute rat slices[38]. The OT-specific stimulation of glutamatergic neurons might modulate GABA-ergic interneurons that produces a generalized elevation of local inhibition, resulting in the reduction of incoming Aδ and/or C primary afferent-mediated sensory transmission. In the deep layer of the spinal dorsal horn, endogenous OT released through PVN stimulation could reduce or prevent the long term potentiation (LTP) in spinal WDR neurons[39]. Subpopulation of OT neurons in parvocellular PVN send their projections exclusively to the deep layers (V, VI, and X) of spinal dorsal horn, suggesting that OT modulates the excitability of WDR neurons[14]. These studies might also explain the increased mechanical/heat thresholds that were observed in the present study.

Besides its anti-nociceptive actions, GO terms related to "inflammatory response", "immune response", "insulin-like

growth factor-1 (IGF-1) binding", and "angiogenesis" were enriched by the chemogenetic activation of OT in the spinal dorsal horn. It is still unclear whether the inflammatory response or immune response directly affected mechanical/heat threshold. However, in the Seltzer model, inflammatory or immune modification by endogenous OT might alleviate hyperalgesia since inflammation of the spinal cord is the main pathology of the neuropathic pain. Also, the results indicated that OT may be involved in remodeling of the injured spinal cord. OT activates nerve growth factor and IGF-1 which play important roles in the healing process of nerve injury[40,41]. In addition, OT administration resulted in accelerating the recovery from sciatic nerve injury in rats[42]. The results of the OTR antagonist intervention on inflammatory pain models may also support this hypothesis.

The activation of endogenous OT alleviated both neuropathic and inflammatory pain including spontaneous nociceptive behaviors. Strikingly, intrathecally administered OTR antagonist (atosiban) abolished the effects of OT in neuropathic pain model, whilst intraperitoneally administered OTR antagonist (L-371,257) ablated the effects of OT in nociceptive pain model. These results suggest that the main site of action of OT is different depending on the type of pain. It should be noted that atosiban is a biased agonist of OTR-Gi pathway rather than an OTR antagonist. Atosiban has been shown to eventually inhibit the function of the neurons or cells that express OTRs. We chose to use atosiban as an OTR antagonist for pragmatic reasons since it has been widely used as an OTR antagonist. Previous studies have reported the analgesic effects of OT on neuropathic pain and nociceptive pain in rodents[43,44]. In humans, however, it is still controversial[45]. The reason for the discrepancy is probably due to OT's short half-life, poor BBB penetration, and lack of specificity to OTRs since OT has similar affinities with V1aR and the Transient Receptor Potential Vanilloid type-1 (TRPV1)[46]. Previous studies have reported that analgesia induced by systemic OT treatment is mediated by the V1aR and TRPV1[5,46]. Other kinds of social behavior such as social interactions induced by OT is also mediated by V1aR[47]. Recently, a OTR agonist with greater specificity and longer half-life could induce a long-lasting reduction in inflammatory pain-induced hyperalgesia symptoms[48]. For the use of OT as an analgesic drug in the clinic, these types of OTR agonists are promising and warrant further investigation.

Although beyond the scope of this study, DRG might be also affected by endogenous OT as OTRs are expressed in afferent neurons of the DRG. OTRs are expressed predominantly in non-peptidergic C-fiber cell bodies in the DRG[49], indicating that humoral OT can penetrate DRG and directly act at the peripheral level of pain structures. It is speculated that direct action of OT on DRG, as well as on anti-inflammatory action, might contribute to its peripheral analgesic effects.

Chemogenetic activation of OT alleviated the swelling of ipsilateral foot pad after formalin injection by inhibiting mast cell degranulation. Previous study has reported that cardiac ischemia/

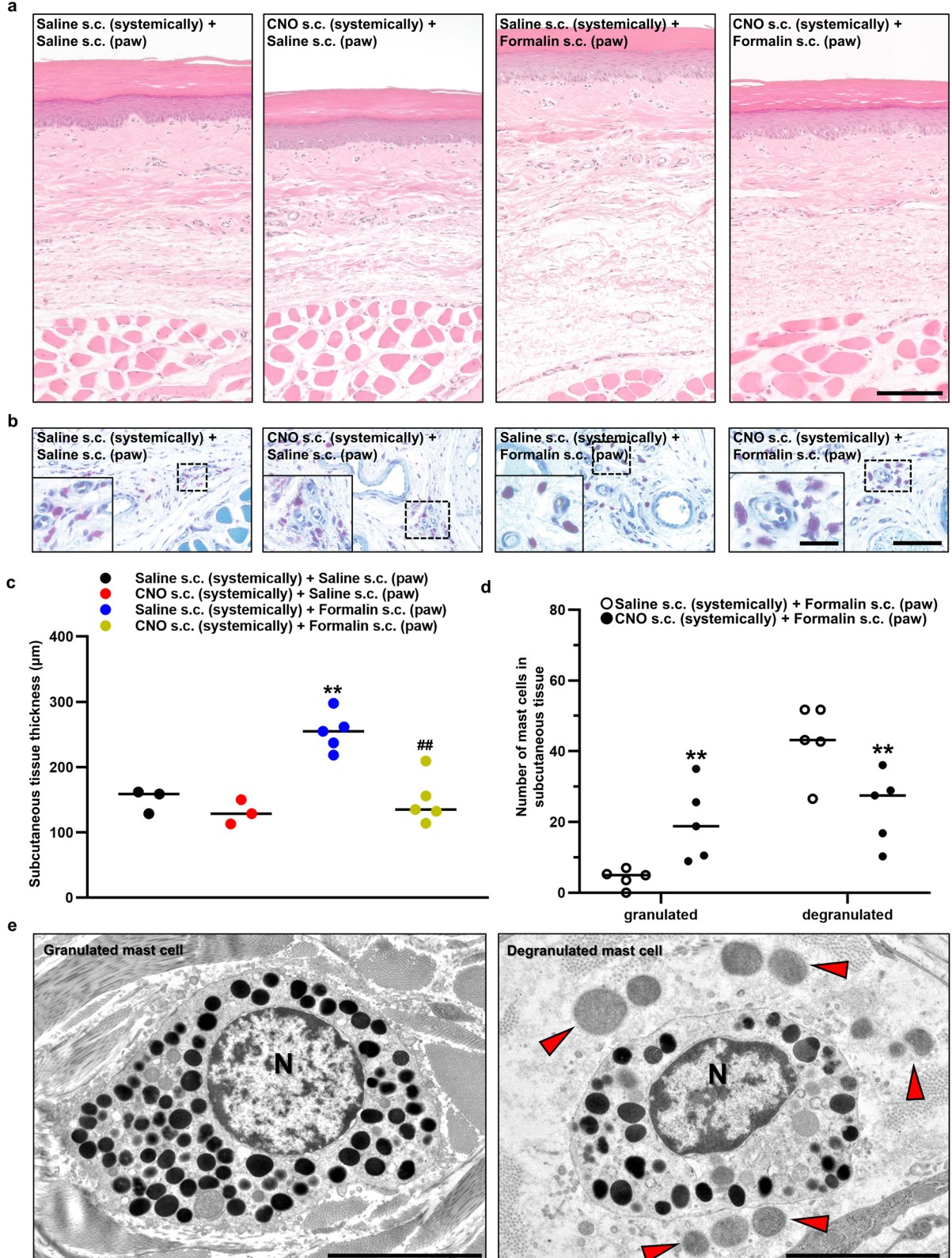

**Fig. 8 OT exerted an anti-inflammatory response by inhibiting mast cell degranulation.** Morphology of hematoxylin and eosin (HE)-stained (**a**) and toluidine blue (TB)-stained (**b**) hind paw after the s.c. injection of saline or 5% formalin (100 μL) into the hind paw. Scale bar in (A), 50 μm. Scale bar in (B), 10 μm and 5 μm (in magnified image). **c** The subcutaneous tissue thickness measured using the HE-stained hind paw slices (n = 6–10 slices from 3 to 5 rats, each). **P < 0.01 vs. Saline s.c. + Saline s.c. ##P < 0.01 vs. Saline s.c. + Formalin s.c. **d** Number of mast cells in subcutaneous tissue either granulated or degranulated was manually counted (n = 10 slices from 5 rats, each). **P < 0.01 vs. Saline s.c. + Formalin s.c. **e** Granulated and degranulated mast cell captured by transmission electron microscope (TEM) after formalin injection. N, nucleus. Red arrow heads indicate the granules that were degranulated from a mast cell. Scale bars, 5 μm. See also Supplementary Fig. 9.

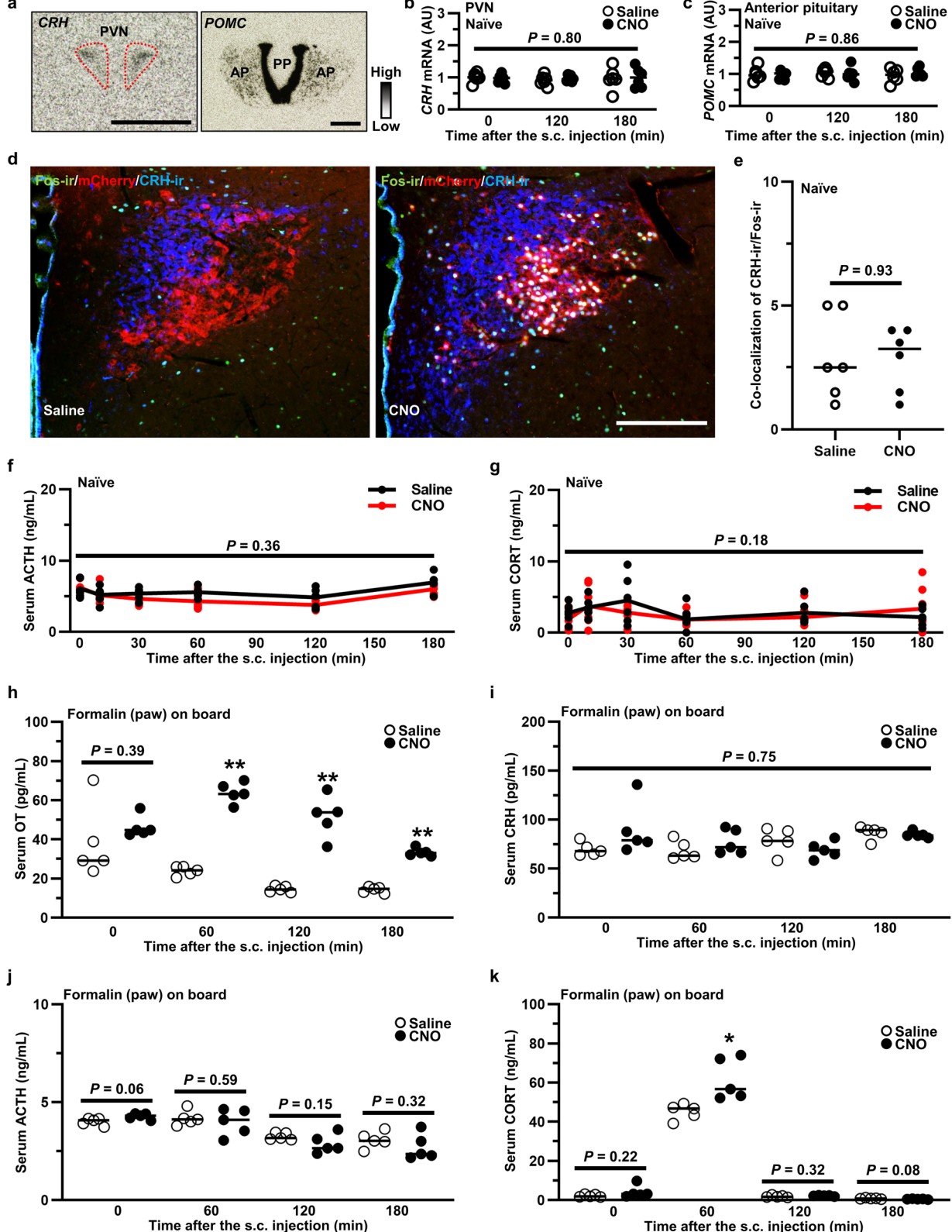

reperfusion injuries were attenuated by inhibition of the degranulation of cardiac mast cells by pretreatment with OT[50]. Petterson et al. have examined the effect of OT on carrageenan-induced inflammation in rat hind paws[51]. They demonstrated that OT administration reduced the edema of the paw with reduced activity of myeloperoxidase. They also suggested that OT's anti-inflammatory effect of OT was comparable to the effect

of the glucocorticoid. Although OTRs are expressed in mast cells, they are also expressed in the fibroblasts that exist closely to mast cells[52]. Peripheral OT might therefore inhibit degranulation of mast cells by modifying the function of the fibroblasts.

Previous studies have shown that OT might modulate the HPA axis[53,54]. OT's effect on the HPA axis probably reflects axonal transport and local regulation of transcription factors in the

**Fig. 9 OT affected hypothalamus–pituitary-adrenal (HPA) axis under pathological condition. a** ISH images of *corticotropin releasing hormone* (*CRH*) in the PVN and *pro-opiomelanocortin* (*POMC*) in the anterior pituitary (AP). Scale bar, 200 μm. PP posterior pituitary. Gene expression of *CRH* in the PVN (**b**) and *POMC* in the AP (**c**) after the s.c. injection of Saline or CNO (1 mg kg⁻¹) ($n = 12$ slices from 6 rats, each). **d** Fos-ir (green), endogenous mCherry (red) and CRH-ir (blue) in the PVN at 120 min after the s.c. injection of Saline or CNO (1 mg kg⁻¹). Scale bar, 200 μm. **e** Number of CRH-ir neurons co-expressed with Fos-ir neurons in the PVN ($n = 12$ slices from 6 rats, each). The serum concentration of adrenocorticotropic hormone (ACTH) (**f**) and corticosterone (CORT) (**g**) after the s.c. injection of Saline or CNO (1 mg kg⁻¹) ($n = 6$ rats in each group at each time point). Serum concentrations of OT (**h**), CRH (**i**), ACTH (**j**), and CORT (**k**) after the s.c. injection of 5% formalin (100 μL) into right hind paw ($n = 5$ rats in each group at each time point). Saline or CNO (1 mg kg⁻¹) was s.c. (systemically) administered at 0 min. *$P < 0.05$; **$P < 0.01$ vs. Saline at the same time point. See also Supplementary Fig. 10.

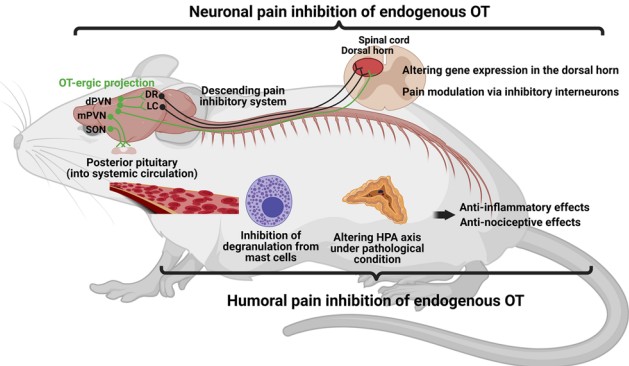

**Fig. 10 The putative mechanism of endogenous OT on pain modulation. A** model describing the putative mechanism of endogenous OT on pain modulation is illustrated schematically. Central OT may exert anti-analgesic effects both directly and indirectly via descending inhibitory system. On the other hand, peripheral OT may alleviate inflammatory response by inhibiting degranulation from mast cells and by modulating HPA axis under pathological condition, resulting in hindering pain transmission. Created with BioRender.com.

neural lobe in the PP[55]. Indeed, we have confirmed that the HPA axis was not affected under naïve condition but was affected after formalin injection. Neumann has reported the effects of OT on HPA axis under physical and psychological stress[56]. OT^PVN neurons exert a tonic inhibition on ACTH secretion possibly via inhibiting CRH neuronal activity under basal state. On the other hand, under the stressed condition, OT^PVN upregulates HPA axis via activating CRH neurons. However, further studies are needed to clarify the effects of OT on the HPA axis as there are still many unknowns.

In the present study, we have succeeded in generating a transgenic rat line, expressing excitatory DREADDs specifically in OT neurons. The development of chemogenetics and optogenetics has revolutionized the field, especially for investigating the role of specific peptides in behavioral changes by enabling activation of specific neurons. Virus-mediated transfection, also known as transduction, has been well-established and commonly used to insert foreign genes into targeted neurons. This technique, however, can induce immunogenicity and cytotoxicity. In addition, the method is not suitable for inserting larger-sized genes[57] and specialized technique is required to achieve consistent expression levels of foreign genes. The transgenic rat model is therefore a greatly refined method of achieving the same goals. Although time consuming in development, the transgene expression is very stable across the generations. An additional advantage is that experiments can be performed in much younger animals, even in pups, since transgenes are expressed from birth; this cannot be achieved by the virus-mediated transfection. The transgenic rats may be further applied for investigating the function of OT in various pathophysiological conditions, such as autoimmune diseases and metabolic syndrome. To date, both optogenetic and chemogenetic methodologies have been applied

to OT neurons. Although most studies have utilized virus-mediated transfection, transgenic rat or mouse lines have also been generated[58–60]. These studies were distinct from ours, as they were investigating more targeted OT-ergic function. On the other hand, our transgenic rats are potentially more suitable for evaluating the physiological function of an individual when all OT neurons are activated. One might even speculate that physiological stimulation may also induce global activation of OT neurons, rather than just selected populations of cells. However, this speculation is currently untested. Of course, the type of genetic model used depends on the specific research question. Certainly, for studies interrogating the neural circuitry, the targeted methodology would be preferable.

A number of studies have demonstrated that OT has anti-nociceptive and anti-inflammatory effects. Our present study using DREADDs provides further support for these conclusions. Although some controversies about OT's role still exist, presumably due to methodological differences employed in different studies, OT, as a candidate analgesic drugs, appears to hold considerable translational promise for the treatment of pain in patients[61]. Since OT is involved in a wide range of behaviors, including social, cognitive, and emotional behaviors as well as modulation of inflammatory processes that might contribute to pain perception, it is quite complicated to dissect its pure analgesic effects from its other effects. Extensive studies are anticipated for a better understanding of OT on anti-nociceptive and anti-inflammatory processes.

## Methods

**Animals and ethics approval**. Non-transgenic and heterozygous transgenic Wistar (CrLj:WI, Japan Charles River, Yokohama, Japan) rats were bred and group-housed ($n = 3$ per cage) under normal laboratory conditions (temperature, 24 ± 1.0 °C; 12/12 h light/dark cycle, lights on at 22.00 h; humidity, 55 ± 5%) with free access to food and tap water at least for 2 weeks. All rats used in the experiments were acclimatized in the reversed light condition for at least 2 weeks. All experiments were commenced in the morning (10.00 h, after lights off), since we assumed that OT reached nadir at that time point, allowing us to assess the effects of activated OT more accurately. All experiments in this study were performed in strict accordance with guidelines on the use and care of laboratory animals as set out by the Physiological Society of Japan and approved by the Ethics Committee of Animal Care and Experimentation, University of Occupational and Environmental Health (approval No. AE21-006).

**Constructs for microinjection**. A chimeric OT-hM3Dq-mCherry BAC clone transgene construct was purified for microinjections. The hM3Dq-mCherry sequence from the hM3Dq-mCherry cassette (Plasmid #44361, Addgene, Cambridge, MA, USA) was used for the transgene[62]. Next, SV40 poly A sequence was framed to the hM3Dq-mCherry sequence. Finally, an hM3Dq-mCherry-SV40 poly A cassette was introduced into the rat OT gene in place of the genomic start codon. Hence, hM3Dq-mCherry should be specifically expressed under the OT promoter in the transgenic rat. Three transgenic founder male rats were identified by Southern blot analysis using genomic tail DNA with a ³²P-labeled mCherry probe. The copy number of the transgene was 10. All founders were bred and F1 rats were screened by PCR analysis of genomic DNA extracted from rats' ear skin. Transgenic founder rats were bred with non-transgenic Wistar rats. F1 heterozygous transgenic rats were screened by PCR analysis of genomic DNA extracted from rats' ear skin.

**Test substances**. CNO (Sigma-Aldrich Japan Co. LLC., Tokyo, Japan) was dissolved in saline (Otsuka Pharmaceutical Co. LTD., Tokyo, Japan)[63]. OTR

antagonist Atosiban (Sigma-Aldrich Japan Co. LLC., Tokyo, Japan) was dissolved in saline and L-371,257 (Tocris Bioscience, Bristol, UK) was dissolved in DMSO[11,64]. Disodium cromoglicate (DSCG) (Tokyo Chemical Industry Co. Ltd., Tokyo, Japan) was dissolved in saline and used as a mast cell stabilizer[65].

**Fluorescent immunohistochemistry**. The rats were deeply anesthetized with i.p. injection of three types of mixed anesthetic agents (in combination with 0.3 mg kg$^{-1}$ of medetomidine, 4.0 mg kg$^{-1}$ of midazolam, and 5.0 mg kg$^{-1}$ of butorphanol). They were transcardially perfused with 0.1 M phosphate buffer (PB) (pH 7.4) containing heparin (1000 U L$^{-1}$), followed by 4% paraformaldehyde in 0.1 M PB. The brains were carefully removed, and a small block that included the hypothalamus was isolated. The blocks were post fixed with 4% paraformaldehyde in 0.1 MPB for 48 h at 4 °C. Then, the tissue was cryoprotected in 20%-(w/v) sucrose in 0.1 M PB for 48 h at 4 °C. Fixed tissue was cut into 30 μm using a microtome (REM-700; Yamato Kohki Industrial Co. Ltd, Saitama, Japan).

The observed nuclei were identified according to the coordinate that given in the rat brain atlas[66]. Information of the primary (Supplementary Table 1) and secondary antibodies (Supplementary Table 2) are summarized in the present study. Sections cut by a microtome were rinsed twice with 0.1 M phosphate-buffered saline (PBS) and washed in 0.1 M PBS (pH 7.6) containing 0.3% Triton X-100 (PBST). They were incubated in a primary antibody solution for 48 h at 4 °C. After being washed twice in 0.1 M PBST, the floating sections were incubated in a secondary antibody solution for 2 h at room temperature. They were then washed twice in PBS for 10 min, and mounted on a slide glass and cover-slipped using a vectashield (Vector Laboratories Co. Ltd., CA, USA). Images scanned by a confocal laser scanning microscopy were reconstructed by using imaging software (ZEN 3.2. blue edition) provided with the LSM880 laser scanning microscope (Carl Zeiss Co. Ltd. Oberkochen, Germany). In addition, images scanned by All-In-One microscopy (BZ-800, Keyence, Osaka, Japan) were used to analyze the percentages of Fos-ir induction.

**Co-localization of hM3Dq-mCherry, OT, VP, CRH, and Fos**. Each captured image by a confocal laser scanning microscopy was printed onto a paper in an expanded size. Subsequently, the printed papers were blinded and endogenous mCherry positive neurons and each protein-ir neurons were manually counted by at least two researchers to avoid skewing the results. We counted two cross sections (four nuclei including right and left) of each nucleus and the results were averaged. To prevent double-counting, we checked the cross mark on the printed paper every time we counted.

**Serum OT and VP concentration (radioimmunoassay (RIA))**. At 0, 10, 30, 60, 120, and 180 min after the s.c. administration of Saline or CNO (1 mg kg$^{-1}$), the rats were decapitated immediately without being anesthetized. The number of rats used in this experiment was 6–7 in each group at each time point (78 rats in total). Trunk blood samples were collected into chilled reaction tubes (Greiner Bio-One Co. Ltd., Kremsmuenster, Austria). Blood samples were centrifuged for 10 min at 4 °C, 1000 × g. After centrifugation, a 1 mL sample of serum was taken for measuring OT and VP concentration.

RIAs for VP, after acetone-ether extraction of plasma, and OT were performed by use of an anti-VP and anti-OT antibodies developed and characterized in our laboratory using methods[67]. The standard curves of VP and OT linear between 0.1–25 pg per tube and 0.4–12.5 pg per tube, respectively. All RIAs were run at two different dilutions in duplicate.

**In situ hybridization (ISH) histochemistry**. Brains and pituitaries were cut into coronal 12 μm sections, and thaw mounted on gelatin/chrome alum-coated slides. The locations of the nuclei including the SON, PVN, DR, LC, and PP were determined according to the coordinates given in the rat brain atlas[66]. $^{35}$S 3′-end-labeled deoxyoligonucleotide complementary to transcripts encoding *OXT*, *AVP*, *CRH*, *POMC*, *TPH2* and *TH* were used. Oligoprobes used in the present study are represented (Supplementary Table 3). Hybridized sections were exposed to autoradiography film (Hyperfilm, Amersham, Bucks, UK) for 12 h for *OXT* probe, 24 h for *AVP* probe, 2 days for *POMC* probes, 1 week for *CRH* probes and 2 weeks for *TPH2* and *TH* probes. The amount of bound probe to mRNA was analyzed in comparison to $^{14}$C-labeled standards (Amersham, Bucks, UK) using image analysis software (NIH Image 1.6.2, W. Rasband, NIH, Bethesda, MD, USA). The obtained results were represented in arbitrary units setting the mean optical density (OD) obtained from control rats[68].

**Retrograde tracer microinjection into the spinal dorsal horn**. All rats were handled for 5 days before the experiment. Retrograde tracer was microinjected into spinal dorsal horn[69]. After rats were deeply anesthetized with i.p. injection of three types of mixed anesthetic agents (in combination with 0.3 mg kg$^{-1}$ of medetomidine, 4.0 mg kg$^{-1}$ of midazolam, and 5.0 mg kg$^{-1}$ of butorphanol), the spinal cord was exposed by performing a laminectomy over one lumbar segment (between L3 and L5). The dura was cut, subsequently a glass micropipette (tip diameter 20–50 μm) was inserted into the dorsal horn of the spinal cord at an angle of 45 degree to the rostrocaudal axis. Green IX Retrobeads® (LUMAFLUOR INC., Durham, NC, USA) were bilaterally injected (10 nL, each) using a calibrated

injection system (Drummond Nanoject, Broomall, PA, USA). Then, the incision was then closed. Animals were allowed to recover from anesthesia in a warmed box before being returned to the holding cages. Seven days after microinjection, they were perfused and fixed. Brains and spinal cords were collected, and cut into 30 μm of coronal brain and axial spinal sections. Images were scanned by BZ-800 (Keyence, Osaka, Japan) to confirm the existence of Reatrobeads® in the dPVN, LC and DR that were retrograded from the spinal dorsal horn.

**Laser microdissection of spinal cord**. All rats were handled for 5 days before the experiment. At 120 min after the s.c. administration of Saline or CNO (1 mg kg$^{-1}$), the rats were decapitated immediately without being anesthetized. Spinal cords were immediately removed, and frozen blocks containing L5 section were prepared using an ultra-low temperature freezer (PINO-600, Sakura Finetek, Osaka, Japan). Then, axial frozen spinal sections (12 μm) were made using a cryostat and mounted on RNase-free slide glasses with foil (Leica, Wetzlar, Germany). Glasses slides were put into 100% ethanol for 1 min to fix spinal sections, thereafter, stained with 0.03 % Toluidin blue for 5 min. Slides were washed with DEPC water for 1 min, then put into 70% ethanol for 1 min to dry. Specimens including laminae I–II and III–VI were collected separately into 0.5 ml PCR tubes using LMD 6 (n = 30 specimens from 3 rats, each) (Leica, Wetzlar, Germany). Finally, TRIzol® (Thermo Fisher Scientific, Waltham, MA, USA) was added into the PCR tube and sent to Genome Biology Lab, Transborder Medical Research Center, University of Tsukuba for RNAseq.

**RNA purification and sequencing**. Each micro-dissected tissue of dorsal horn was treated with TRIzol® reagent (Thermo Fisher Scientific, Waltham, MA, USA). Purified total RNA was re-suspended in $H_2O$. Total RNA was quantified using a Nanodrop spectrophotometer (Thermo Fisher Scientific, Waltham, MA, USA) and an Agilent Bioanalyzer RNA 6000 Pico Kit (5067-1513) (Agilent Technologies Japan, Ltd., Tokyo, Japan) for RNA quality control (QC). For library synthesis, 50 ng total RNA was used for rRNA-depletion and directional library synthesis with SMARTer Stranded RNA-Seq Kit (TakaraBio, Shiga, Japan). Library QC was performed using TapeStation High-sensitivity D1000 ScreenTape (5067-5584, Agilent Technologies Japan, Ltd., Tokyo, Japan). Then, each sequencing was performed using an Illumina NextSeq500 (Illumina, Inc., San Diego, CA, USA) with a high-output kit in paired-end reads (v2, 2 × 36), according to the manufacturer's instructions. Raw RNA sequencing data are available at Gene Expression Omnibus (GEO) in the National Center for Biotechnology Information (NCBI) (accession number GSE210528).

**RNAseq data analysis**. FASTQ files were imported to CLC Genomics Workbench (CLC GW, v10.1.1, Qiagen, Hilden, Germany). Sequence reads were mapped to rat reference (rn6) and quantified for 32,623 genes. Mapped reads were visualized by exporting BAM files, converting into bedgraph files, and uploading to UCSC Genome Browser (https://genome.ucsc.edu/). Box plot and principal component (PCA) plot were produced using raw read counts in CLC GW. To identify differentially expressed genes, total read counts were analyzed by Empirical Analysis of DGE tool in CLC GW. Altered genes were filtered by |Fold Change (FC)| and p values. Raw total counts were normalized by quantile method. After addition of 1 to avoid 0, normalized counts were Log$_2$-converted, and imported to Morpheus (https://software.broadinstitute.org/morpheus/) to draw heatmap. Volcano plot was produced with |FC| and p value exported from CLC GW. GO analysis was performed using Metascape webtool[70]. GraphPad Prism 9 (GraphPad Software, San Diego, CA, USA) was used to produce plots for results exported from CLC GW and Metascape.

**von Frey and hot plate tests**. All rats were handled for 5 days before each experiment. Before the tests, all rats were put into the device and were acclimatized to an acrylic cage on an elevated mesh floor at least 30 min before the test. The manual von Frey test was performed using calibrated von Frey filaments (North Coast Medical, Gilroy, CA, USA) to evaluate the mechanical sensitivity. Repetitive measurements were performed using von Frey filaments ranging from 0.25 to 20.0 g to the plantar surface of the ipsilateral foot using ascending stimulus method[71]. The stimulus was applied five times over a three seconds time interval, and the weakest force (g) to induce a paw withdrawal was regarded as the mechanical nociceptive threshold. Averaged values were calculated in each animal at each time point. This measurement method (5 times von Frey fibers in 3 s) may induce temporal summation.

A hot plate test was performed to measure the heat sensitivity. Rats were placed gently on the heated plate (52.5 °C), and the latency of nocifensive behavior (licking, lifting, shaking or jump) was measured, then the mean value was calculated and analyzed[72].

**Formalin test**. Rats were briefly anaesthetized with inhalation of sevoflurane for 2–3 min in a glass chamber. After disappearance of spontaneous movement with preservation of the deep spontaneous respiration, blink and pinnae reflexes, 100 μl of 5% formalin was injected subcutaneously into the planter surface of the right hind paw with a 26 gauge needle[73]. Immediately after the formalin injection, the animal was placed in an open Plexiglas box (10 × 20 × 24 cm). Time point 0 was

defined as the commencement of licking their formalin-injected limb in Saline-treated rats. Formalin-induced pain evokes three main behavioral responses: licking, tonic flexion and paw jerk. They were recorded for 60 min after the formalin injection. The total time of licking the injected hind paw (at 5 min interval for 60 min) were analyzed. Pain induced by formalin in rodents has two phases which reflect different pathological processes. After the formalin injection, animals show early or acute painful responses (0–7 min) which imitate the direct activation of nociceptors, then, attenuation or quiescent of nociceptive responses in an interphase is observed, followed by a long-lasting period of nociceptive behaviors which might last for more than 45 min. According to the previous study[72], we defined the 1st and 2nd phase as 0–10 and 10–60 min, respectively. The results were blinded and analyzed randomly in duplicate by at least two researchers to avoid bias.

**Carrageenan knee arthritis model**. The rats were anesthetized with inhalation of sevoflurane for 2–3 min in a glass chamber. An i.a. injection of 0.1 mL of 3% λ-carrageenan (Sigma, St. Louis, MO, USA) which was dissolved in 0.9% NaCl was administered into the right hind knee joint using 25-gauge injection needles[29].

**Measurement of joint swelling**. To assess joint swelling induced by carrageenan i.a. injections, the diameters of the right and left knee joints were measured using digital calipers before the i.a. injection and at 3, 6, and 12 h after the i.a. injection on the same day. The distance between the lateral and medial collateral ligament was defined as the knee joint diameter[74]. Averaged diameter for each group at each time point was calculated and analyzed.

**Seltzer model**. After deeply anesthetized with i.p. injection of three types of mixed anesthetic agents (in combination with 0.3 mg kg$^{-1}$ of medetomidine, 4.0 mg kg$^{-1}$ of midazolam, and 5.0 mg kg$^{-1}$ of butorphanol), the one-third to half diameter of the right sciatic nerve was ligated with 6-0 silk suture after exposure[27]. The rats that showed drop-foot were omitted from the analysis.

**OTR antagonist treatment**. OTR antagonist was administered via i.t. and/or i.p. injection. Rats were briefly anesthetized with sevoflurane. Twelve-μL of Atosiban dissolved in saline (1 μg μL$^{-1}$) was intrathecally injected by acute puncture between L3 and L4 vertebrae of the spine using a 30 gauge needle[43]. Needle insertion was confirmed by a brief tail flick. L-371,257 dissolved in DMSO (10 mg mL$^{-1}$) were intraperitoneally injected[64].

**Pathological examination after formalin test**. Right feet of rats, which were used for the formalin test, were amputated, and kept in 4% formalin. Then, fixed feet were cut into 6 μm sections and stained with hematoxylin and eosin (HE) and toluidine blue (TB) to assess morphological changes.

**Transmission electron microscope observation**. Tissues were fixed overnight in 2.0% paraformaldehyde, 2.5% glutaraldehyde in 0.1 M phosphate buffer at 4 °C, and post fixed in 1% osmium tetroxide in 0.1 M phosphate buffer for 2 h at 4 °C. Specimens were dehydrated and embedded in epoxy resin. Semithin sections were stained with toluidine blue, and observed under light microscope. Thin sections were stained with uranyl acetate and lead citrate, and observed under transmission electron microscope at 80 kV (1400Plus, JEOL, Tokyo, Japan).

**Measurement of serum CRH, ACTH, and CORT (ELISA)**. Plasma concentrations of CRH (YK131 Mouse/Rat CRF-HS ELISA kit, Yanaihara Institute Inc., Fujinomiya, Japan), ACTH (ACTH Rat, Mouse EIA kit, PHOENIX PHARMACEUTICALS, INC., CA, USA), and CORT (Corticosterone ELISA kit, Cayman Chem., MI, USA) were analyzed by ELISA. Procedures were implemented by a protocol attached each ELISA kit. All results were duplicated, and the averaged value in each sample was calculated.

**Restraint stress**. The transgenic rats were pretreated with Saline or CNO (1 mg kg$^{-1}$) at 30 min prior to the restraint stress. Restraint stress was induced by taping all 4 limbs of the rat to metal mounts attached to a wooden board[75]. The immobilization lasted for 30 min as an acute stress stimulus. Rats were killed by decapitation without being anesthetized to collect the trunk blood immediately after the end of the immobilization.

**Statistics and reproducibility**. The mean ± standard error of the mean (SEM) was calculated from the results. All data were analyzed by student $t$ test, one-way ANOVA, or two-way ANOVA followed by a Bonferroni-type adjustment for multiple comparisons using R software. Statistical significance was set at $P < 0.05$.

**Reporting summary**. Further information on research design is available in the Nature Research Reporting Summary linked to this article.

## Data availability
The materials, including OT-hM3Dq-mCherry transgenic rats, are available from the corresponding authors, upon reasonable request. The data that support the findings of this study are available from the corresponding authors, upon reasonable request. Raw RNA sequencing data are available at Gene Expression Omnibus (GEO) in the National Center for Biotechnology Information (NCBI) (accession number GSE210528).

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

## Acknowledgements

This paper was supported by a Grant-in-Aid for Scientific Research (C) (21K06779) KAKENHI for M.Y., a Grant-in-Aid for Young Scientists (20K22749) KAKENHI for S.S., and a Grant-in-Aid for Scientific Research (B) (17H04027) KAKENHI for Y.U. from the Ministry of Education, Culture, Sports, Science, and Technology (MEXT), Japan; and a University of Occupational and Environmental Health (UOEH) Grant-in-Aid for Priority Research in the Field of Occupational Medicine (2021, 2022) for M.Y. from the UOEH, Japan.

## Author contributions

Conceptualization: M.Y. and Y.U.; Data acquisition: H.N., M.Y., M.S., K.S., S.S., K.N., K.B., N.I., T.M., Y.N., R.B., T.O., T.H., H.M., and Y.Y. Data analysis and interpretation: H.N., M.Y., Y.M., R.B., H.M., Y.Y., M.K., A.S., M.M., B.C.C., S.L., and Y.U. Preparation of the draft and figures: H.N., M.Y., B.C.C., and M.M. Final approval: M.Y. and Y.U. All authors have approved the final version of the paper and agreed to be accountable for all aspects of the work. All authors were designated as authors qualified for authorship, and all those who qualify for authorship are listed.

## Competing interests

The authors declare no competing interests.
