## [Peer Review File · Communications Biology]

Reviewers' comments:

Reviewer #1 (Remarks to the Author):

This study by Haruki Nishimura, Mitsuhiro Yoshimura and colleagues describes the anti-nociceptive and anti-inflammatory effect of endogenous OT activity in rats. The authors generated a transgenic rat line expressing the excitatory DREADD hM3Dq coupled to mCherry in OT neurons via exclusive expression under the control of the OT promoter. They demonstrated the functionality of the transgenic line after CNO injection (chemogenetic activation) via Fos staining, in situ hybridization, and analysis of serum OT in both the PVN and SON. A mechanical and thermal anti-nociceptive effect was shown via von Frey and hot plate test after chemogenetic activation of OT neurons. Additionally, the authors showed that the expression of genes that are involved in inflammatory and immune responses and that are predominantly expressed in GABAergic neurons were significantly altered. While authors described an anti-nociceptive effect of endogenous OT in the neuropathic pain model mediated via the neuronal pathway utilizing i.p. and i.t. injections of an OTR antagonist, they showed an anti-nociceptive effect mediated via the humoral pathway in the inflammatory pain model (formalin injection and carrageenan knee arthritis). Finally, the authors showed that degranulation of subcutaneous mast cells was inhibited by endogenous activation, while no alteration of CRH or POMC gene expression or Fos-activity was found in naïve animals, concluding that the HPA axis is not affected by this system.

Comments

In general, this work is structured in a comprehensive way, with figures supporting the main messages of the manuscript. The order in which the results are presented is coherent. The titles of sections and figures accurately describe the content and briefly summarize the findings, without overrating or hyping the actual results. Especially the creation of a new transgenic rat line is of high importance to researchers in the field, and findings about differential functions in the periphery vs central nervous system are relevant for future research.

However, some major and minor comments about the content are listed below, which would improve the message of the manuscript if addressed by the authors before publication.

1. Regarding the novelty of the findings, as authors mentioned the anti-inflammatory effect (Pettersson et al., 2001) and changes in von Frey sensitivity (MiguelCondés-Lara et al., 2005) after exogenous OT application were known before. However, the authors claim to be the first to endogenously activate exclusively OT neurons in PVN and SON and investigate pain in this system (p4 l83-85). While this is the first transgenic rat line (to my knowledge) to perform these experiments, it has actually been already performed and published before. With both optogenetic and chemogenetic activation targeted exclusively to OT neurons via rAAVs, Eliava et al. (2016) showed that endogenous OT release by fibers originating from PVN directly on WDR neurons inhibits sensory processing and produces analgesia in an inflammatory pain model, also measured by mechanical threshold and withdrawal. This study also already backlabeled OT neurons from the PVN to L5, as done in this manuscript.

2. Authors hypothesized that the involvement of the HPA axis (via CRH-ir and Fos-ir colocalization or CRH and POMC gene expression) was not found, because experiments were performed only in naïve animals. As authors stated themselves, it would be really interesting and add a lot to the manuscript if they could test this in non-naïve (i.e. inflammatory or neuropathic pain model) animals, and include it already in this study.

3. Please also provide p values of non-significant tests

4. Please indicate clearer whether n is the number of animals or slices analysed.

5. In this work, authors differentiated between parvocellular and magnocellular neurons solely based on anatomical localisation. It could be more precise to determine cell-types based on their projection to the posterior pituitary via a retrograde tracer (e.g. fluorogold) for a more reliable differentiation.

6. Most data is shown both as raw values and additionally as delta (to timepoint 0). This is in most cases redundant and does not add any additional beneficial information.

7. The numbers of %colocalization or %Fos expression (mCherry & hM3Dq expression verification, p5 l101ff) should be given in the text (and not just depicted in figures). Does the from literature referred

to 3-5% VP/OT overlap expression match the counting in the authors' experiments (e.g. Fig. 1F, right)? In this same section, the shown data appears to be clearly significant, but why was no statistical analysis performed (e.g. "Fos expression [...] was robustly increased", instead of "significantly ($p < xxx$)")?

8. Do the shown %Fos expressions refer to the percentage of mCherry neurons expressing Fos, or the other way around? On a first glance, it looks like many Fos-positive (green) neurons are not mCherry labelled. Could you discuss this? (e.g. activation of OT neurons also activates "down-stream" neurons in the PVN and SON?)

9. p5 l 127ff mCherry expression in PP seems very unspecific. Fig. S2 D is zoom of which region exactly? The mCherry expression does not look like fibres projecting to PP. How do you conclude expression in terminals?

10. It is great that the unexpected significant effect in WT controls is reported. How would you explain this effect, and how does it not invalidate the findings in the transgenic rat line?

11. Briefly describe (as you did for Fos and PAX2) what tyrosin-hydroxylase and tryptophan hydroxylase are a measure for. Also, why is in Supp. Fig. 3 LC only TH-ir shown, and in DR TPH-ir, while in Supp. Fig. 4 TPH-ir in LC and TH-ir in DR is shown?

12. Along the same lines, why is there a representative image of TH and Fos colocalisation of only DR, and only a counting and ISH image in LC?

13. How do you explain that von Frey and hot plate test (mechanical vs thermal pain) have their only significant anti-nociceptive effect at different timepoints (30 vs 60 min)? Please discuss this in the manuscript.

14. While the difference between OTR a i.p. + vehicle (blue) and OTR a i.p. + i.t. (yellow) in the inflammatory pain model is not significant based on your figures (as compared in Fig. 6 H,I), there seems to be a clear difference between the two at 20-40 minutes? How do you explain this?

15. In the hot plate/von Frey test ablation by OTR antagonist completely alleviates the effect back to the baseline behaviour observed after saline injection. Why is this not the case in the inflammatory pain model in the first 20 minutes (Fig. 6 E, 6, black vs blue and yellow line)?

16. If animals are briefly anesthetized for formalin injection (probably with a variance between animals regarding the "deepness" of anesthesia?) how can you start behavioural readout (e.g. licking) at timepoint 0?

17. Please mention that OT does not cross the BBB, therefore making the differentiation into neuronal/humoral after i.p. and i.t. injection possible.

18. Please also provide the number of counted cells and not only their percentage (dpPVN vs mPVN). Particularly for dpPVN counting of %Fos colocalization seems to be based on very little cells?

19. How/why did you define the first and second phase of licking behaviour etc. as 0-10 and 10-60 minutes?

20. In figure 1I, the legend for PVN is missing (in accordance to Fig. 1F)

21. The conclusion "OT may be involved in the remodeling of the injured spinal cord" should be explained.

22. p18 l595 Mice?

23. supp. fig. 7D Do you have an image with rats in the same position? the CNO animal is angled which makes comparison difficult

24. Fig. 7 A-C Please indicate injection site of saline (it's clear for CNO and formalin, but for saline it would be nice to have "saline s.c. (in paw)" added to the legend)

Spelling/grammar

The manuscript should be checked for correct use of articles (a/an, the) and punctuation (too many/little commas). Examples:

- p3 l46 "THE descending pain inhibitory system"
- p3 l51 "of AN OT receptor antagonist"
- p3 l52 "altering THE hyphothalmus-pituitary-adrenal axis"
- p4 l82 "involved in pain modulation"

Please change direct referring to figures to be consistently: "in the figure (Fig. X)" vs "in figure X"

Typos and rephrasing

- p3 l48ff the sentence should be rewritten to be more clearly
- p4 l64 "neuronal populationS"
- p4 l66 replace "Whilst" with e.g. "On the other hand" or even "Meanwhile"
- p4 l75ff make sentence clearer with structuring, e.g. "on the one hand"/"on the other hand" or "firstly"/"secondly"
- p4 l76 "in the periphery"
- p4 l80 "stress-conditions / stressful conditions"
- p4 l91f rephrase "via centrally and peripherally using the transgenic rats." to clarify
- p5 l99f What is the meaning of this sentence? Was the line "established" after the following described verification of expression?
- p5 l125f Double-negative. "Consistently, neither serum OT nor VP remained unchanged after the s.c. injection of CNO in WT rats" -> means that both changed, which is not the case. "Consistently, both serum OT and VP remained unchanged after the s.c. injection of CNO in WT rats" or "Consistently, neither serum OT nor VP changed after the s.c. injection of CNO in WT rats"
- p5 l141 replace "Whilst", see above
- p6 l145 "OT altered anti-nociceptive behaviour via descending pain inhibitory system" should be rephrased. What is "anti-nociceptive behaviour"? It is being used synonymously with "nociceptive behaviour" throughout the manuscript.
- p6 l146 "were carried out"
- S Figure 3 heading "did not ... neither ... nor" is over-negation and grammatically incorrect
- p6 l158 "were arisen" replace by "did arise"
- p7 l169 add comma for clarification "(TPH)-ir neurons, Fos-ir neurons, and TPH-ir neurons co-expressed with Fos-ir..."
- p7 l188 "... we speculated there to be direct..."
- Supp. Fig.4D "PVN" split into second line
- p7 l199 remove point after quality
- p8 l216 "are rapidly developed, and continue for a substantial amount of time"
- p8 l217 "were promptly"
- p8 l219 "to exclude unwanted effects of surgical intervention, the experiment was performed"
- p8 l225 missing substantive "effects of OT on neuropathic pain were ascribed to the neuronal or humoral targets of OT"
- l230 double negation, see above
- l231 "both" confusing in this context, better "whilst combined i.p. plus i.t. injection"
- Fig 5. legend l924f "Saline or CNO after pretreatment with either an OTR antagonist or vehicle..."
- p8 l245 "Total licking time"
- p8 l1246 "decreased in animals pretreated with CNO compared to SaLine"
- p8 l245f and p9 l251 "analysed every 5 min" would be better "analysed in 5 min intervals or bins"
- p9 l252 "vehicle"
- p9 l285 please rephrase "crucial confrontations"
- p9 l287f double negation, see above
- p10 l304 "elicit" wrong word in this context
- p10 l314 "in the lateral wing of the DR which is a stress-sensitive region"
- p11 l350 "These results suggest that..."
- p12 l374 "mast cells by modifying"
- p16 l 511 "brains and pituitaries"

Reviewer #2 (Remarks to the Author):

In this manuscript, Nishimura et al. present a new transgenic rat line (DRADD-Gq in OT neurons). Taking advantage of this new tool, they activated OT neurons and observed strong anti-nociceptive, anti-hyperalgesic and anti-inflammatory effects. This is an impressive amount of work. The OT-hM3Dq-mCherry rat line is very convincing, with proof of principle made from different techniques, from anatomical observations to circulating OT dosages. I am convinced it will be an excellent tool for

future studies. The numerous tests performed on nociception and pain models are convincing and for the most part well executed. The whole section on anti-inflammation is particularly interesting, as it is rare in the literature to find those aspects analyzed. I have no doubt regarding the interest of the research reported in this manuscript, but several points need to be improved before publication:

Majors:

- A confusion exist in this paper regarding the terminology. Anti-nociception refers to a decrease (or abolition) of normal nociception (Figure 2) while anti-hyperalgesia refers to a decrease in hyperalgesia existing in a pain model (Figure 5). Authors should carefully review this. The definitions provided by the IASP might help (See DOI: 10.1097/j.pain.0000000000001939).

- Important references are missing, such as Eliava et al., Neuron, 2016. Here, authors used similar approach (DREADD and optogenetics) to decipher one of the putative endogenous OT analgesic pathways. A new (apparently not yet peer reviewed) study from the same labs is now online (biorxiv) regarding a new OT pathway toward periaqueductal grey. These make the statement at the end of introduction, p4, incorrect "Direct effects of endogenous OT on pain pathways, however, have not been clarified". While it does not affect the interest of Nishimura et al manuscript, I strongly suggest authors to modify this statement. It is particularly important for the authors to compare their results to other studies involving endogenous oxytocin. This can also be used to enrich the discussion part when approaching the role of OT in spinal cord.

- Page 6, authors say "DR and LC, the nuclei that are involved in the descending pain inhibitory system". I agree those are part of the descending controls. However, many other structures are involved, even if only considering structures in which OT has been suggested to act, such as PAG, RVM, PB, Amy; this has been reviewed several times (e.g. DOI: 10.1007/7854_2017_14). Thus, authors should thus modify the sentence for "DR and LC, two nuclei that are involved in the descending pain inhibitory system". The discussion should also be slightly reviewed considering it, as for now a naive reader could understand that only DR and LC are involved in descending pain controls, while it is wrong.

- Figure 5. Do authors measured the effect of DREADD activation of OT neurons in Sham animals as well? I failed to find this information. Please include the results, as it is a mandatory control for any neuropathic pain model.

- Formalin test: as far as I know, formalin test is used as a nociceptive test. This is not a classical inflammation-induced pain sensitization, as are carrageenan or CFA models. Formalin injection induces a short lasting "spontaneous" nociceptive behavior, while carrageenan/CFA injections induce an inflammation-related hyperalgesia. Authors should consider it while revising their manuscript, in both description of the text and discussion.

- In discussion, authors mention the practical use of OT in clinics. However, three main parameters are limiting its use there: first, its very short half life (5-15min, depending on the compartment), second, its very poor BBB passage, third, its ability to bind V1a, V1b and V2 receptors, with foreseen dangerous side effects. Therefore, chemists are now looking for more specific OTR agonists with longer half life. To my knowledge, only one is currently under investigation for pain (doi: 10.1038/s41598-020-59929-w). This might be mentioned in the discussion.

- While the discussion is interesting, and I really appreciate that author don't overestimate their results, they should add a paragraph regarding the new rat line: it is a major achievement in the field, and highlighting the specificity of expression and the further use for of this tool might be of interest. I believe it is a strong tool, which should not be overlooked.

- Please mention in the text and figure legend the concentrations / quantities used for all molecules.

Minors:

- The OTR antagonist used (L368) (please mention it in the core of the manuscript, not only in methods, for clarity purpose) is known to pass through the BBB. 12ul at 1ug/ul was injected i.t. but how much i.p.? Is it possible that the difference observed between i.t. and i.p. injections is due to a difference in doses?
- Figure S2C, It would be nice if authors additionally provide a low magnitude picture of the posterior pituitary.
- Line 35, "mPVN" refers to medial paraventricular nucleus. It should be defined, as otherwise one might believe it is magnocellular PVN.
- Line 96, I guess "BAC" should be defined. Same for "FIHC" line 116
- Line 99, figure appeal seems to be wrong, 1B should be 1B-C. There are other small mistakes of that order in the manuscript that should be carefully reviewed.
- Line 132, the official nomenclature for oxytocin gene is OXT. For vasopressin, this is AVP.
- Line 141, VP must be defined
- Line 204, reference in wrong format (Haring et al., 2018)
- Page 8, please briefly describe Seltzer model of neuropathic pain in the core of the text, for clarity purpose.
- Knee inflammation: please mention in the text (not only mat and meth) the model used.
- Some typo mistakes are present within the manuscript (e.g. food should be foot in page 9)

Reviewer #3 (Remarks to the Author):

This manuscript by Nishimura and colleagues describes a comprehensive set of experiments using a DREADD approach to clarify aspects of the analgesic and anti-inflammatory actions of oxytocin. In general, I find the data pretty convincing, and clarification of oxytocin's analgesic actions was sorely needed. I have some critiques of the write-up and data presentation, though.

The major problem with the text (aside from the English, which will need to be further edited) is that the authors never really set up what was unclear about the role of oxytocin in pain. They give a reasonable introduction to oxytocin anatomy, and pain anatomy, and then simply cite a few papers suggesting that oxytocin is "involved in pain modulation and exerts anti-nociceptive effects". But the situation is rather more complex than this, and one complication that is mentioned absolutely nowhere in the paper is the findings of Schorscher-Petcu and colleagues in 2010 suggesting that oxytocin analgesia is entirely due to activation of vasopressin-1A receptors. This paper obviously comes to a very different conclusion. Is it a rat/mouse difference? I would have expected much of the discussion to centre around this particular discrepancy, but that paper isn't even mentioned.

I'm somewhat worried about the behavioural pain testing, which based on the methods section is not a core competency of this group. The hot-plate test description is almost completely incomprehensible, as the dependent measure is described as "withdrawal". What does that mean? Accepted dependent measures on the hot-plate test (which probably should not have been performed

twice, because rodents learn escape behaviours with repeated exposure) include hind paw licking, lifting, shaking/fluttering. Were these all measured? For the von Frey test, the psychophysical approach was not stated. Were filaments applied in ascending order? Two stimuli per second is WAY more frequent than von Frey filaments should be applied; this kind of paradigm will produce temporal summation. This all being said, the findings are so robust that even though I think these assays were performed poorly, the findings are probably accurate nonetheless. But clarifications are needed to the methods.

I have absolutely no idea why RNAseq was performed here. In my mind this is a classic example of performing a technique because it exists instead of because it answers a question you need answered. What conclusion do the authors come to about the hundreds of regulated genes? That oxytocin modulates pain both directly and indirectly, that inflammatory/immune response might be involved, and that "many" of the genes are expressed in GABAergic neurons. None of these hypotheses was followed up directly.

The figures are dense, not high-enough resolution, and do not conform to the increasingly universal expectation that individual data points will be shown.

Some other suggestions for improvement include:

1. If "the dorsal part of the PVN" really needs to be abbreviated, it should be "dPVN", not "dpPVN".
2. Why were two completely different OTR antagonists used in the i.p. versus i.t. injections? There might be a reason related to permeability or solubility, but this should be defended.
3. The mast cell hypothesis would be strengthened greatly by the use of a mast cell stabilizer, instead of trying to rule out the alternative via HPA axis gene expression.
4. Why do Figures 5 and 6 variously say "OTR antagonist" versus "OXTR antagonist"?
5. It should be made clearer to the reader that in fig. 5F-G and Fig. 6G-I that CNO is on-board as well as the other drugs.

We are grateful to the reviewers and editor for their rigorous and constructive review of our manuscript. Their careful review helped us to improve our manuscript for better readability. Our answers to the individual comments are enclosed.

Reviewer #1 (Remarks to the Author):

- 1. Regarding the novelty of the findings, as authors mentioned the anti-inflammatory effect (Petersson et al., 2001) and changes in von Frey sensitivity (MiguelCondés-Lara et al., 2005) after exogenous OT application were known before. However, the authors claim to be the first to endogenously activate exclusively OT neurons in PVN and SON and investigate pain in this system (p4 183-85). While this is the first transgenic rat line (to my knowledge) to perform these experiments, it has actually been already performed and published before. With both optogenetic and chemogenetic activation targeted exclusively to OT neurons via rAAVs, Eliava et al. (2016) showed that endogenous OT release by fibers originating from PVN directly on WDR neurons inhibits sensory processing and produces analgesia in an inflammatory pain model, also measured by mechanical threshold and withdrawal. This study also already backlabeled OT neurons from the PVN to L5, as done in this manuscript.**

Thank you very much for your indication. We are very sorry for missing important references. Eliava et al. demonstrated that selective parvocellular OT activation inhibited sensory processing via wide dynamic range (WDR) neurons, which resulted in analgesia in an inflammatory pain model in mice (Eliava. *et al.*, *Neuron*, 2016). They also found that parvocellular OT neurons activated magnocellular OT neurons in the SON and increased OT release. Generally, viral transfection is used for exogenous gene expression, however, this method is fraught with technical challenges. In the present study, as an experimental refinement, we have generated a transgenic rat line that expresses human muscarinic acetylcholine receptor (hM3Dq, excitatory DREADDs) and mCherry exclusively in OT neurons. We have revised Introduction accordingly. Please see our revised Introduction on pages 4 and 5.

- 2. Authors hypothesized that the involvement of the HPA axis (via CRH-ir and Fos-ir colocalization or CRH and POMC gene expression) was not found, because experiments were performed only in naïve animals. As authors stated themselves, it would be really interesting and add a lot to the manuscript if they could test this in non-naïve (i.e. inflammatory or neuropathic pain model) animals, and include it already in this study.**

Thank you very much for your constructive opinions. We performed additional experiments to explore the HPA axis under pathological pain conditions. Immediately after the 5% formalin injection (100 μ L) into the right hind paw, Saline or CNO (1 mg kg⁻¹) was s.c. injected, then trunk blood samples were taken by decapitation. Serum OT, CRH, adrenocorticotrophic hormone (ACTH), and CORT were measured. OT elevation after the s.c. injection of CNO (1 mg kg⁻¹) was confirmed (Fig. 9H). Although serum CRH and ACTH were not altered, CORT was significantly increased at 1 h after the s.c. injection of CNO (1 mg kg⁻¹) (Fig. 9I-K). In addition, restraint stress was performed to explore the modulative effects of OT on HPA axis. Saline or CNO was s.c. injected at 30 min prior to the restraint stress, then rats were immobilized for 30 min. Trunk blood samples were collected just after the end of their immobilization. Although restraint stress itself caused significant elevation of OT, the additional effects of CNO on serum OT was also observed (Supple. Fig. 10). However, no alteration of HPA axis was observed after acute restraint stress (Supple. Fig. 10), indicating that endogenous OT may play differential roles in modulating the HPA axis, depending on the type or duration of stress. Our additional experiments are shown in Fig. 9 and Supple. Fig. 10. Please see the Results 9 *OT affected hypothalamus-pituitary-adrenal (HPA) axis under pathological condition* on page 11. We have also revised the Discussion accordingly.

3. Please also provide p values of non-significant tests.

Thank you for your indication. We have provided *p* values of non-significant tests.

4. Please indicate clearer whether n is the number of animals or slices analysed.

Thank you for your indication. We have revised the Result and Figure legend to indicate the “n” clearer.

5. In this work, authors differentiated between parvocellular and magnocellular neurons solely based on anatomical localisation. It could be more precise to determine cell-types based on their projection to the posterior pituitary via a retrograde tracer (e.g. fluorogold) for a more reliable differentiation.

Thank you for your suggestion. Although parvocellular and magnocellular PVN OT neurons were anatomically defined, 90-95% of OT neurons were activated after chemogenetic activation in both divisions. In the present study, we intended to describe that both parvocellular and magnocellular OT neurons were activated, rather than showing those individually. However, assessing accurate rate of neuronal activation, both in parvocellular and magnocellular PVN OT neurons, would give us beneficial information. We would like to investigate these in a future study.

- 6. Most data is shown both as raw values and additionally as delta (to timepoint 0). This is in most cases redundant and does not add any additional beneficial information.**

Thank you for your suggestion. We have deleted the delta data which we thought to be redundant.

- 7. The numbers of %colocalization or %Fos expression (mCherry & hM3Dq expression verification, p5 l101ff) should be given in the text (and not just depicted in figures). Does the from literature referred to 3-5% VP/OT overlap expression match the counting in the authors' experiments (e.g. Fig. 1F, right)? In this same section, the shown data appears to be clearly significant, but why was no statistical analysis performed (e.g. "Fos expression [...] was robustly increased", instead of "significantly ($p < xxx$)")?**

Thank you very much for your indication. We have added % colocalization and % Fos expression in the main text. In addition, statistical analysis was also performed and described in the figure. Please see Fig. 1. Please also see the Results 1 *Generation of the OT-hM3Dq-mCherry transgenic rat line* on page 5.

- 8. Do the shown %Fos expressions refer to the percentage of mCherry neurons expressing Fos, or the other way around? On a first glance, it looks like many Fos-positive (green) neurons are not mCherry labelled. Could you discuss this? (e.g. activation of OT neurons also activates "down-stream" neurons in the PVN and SON?)**

Yes, correct. Most of the Fos-ir neurons were mCherry positive neurons. Low resolution of the images was one of the reasons since we had to submit our manuscript as one PDF file. These can be distinguished in our original figures, which would be submitted if the manuscript will be accepted. We will upload the Figures as individual PDF files in the revised version.

- 9. p5 l 127ff mCherry expression in PP seems very unspecific. Fig. S2 D is zoom of which region exactly? The mCherry expression does not look like fibres projecting to PP. How do you conclude expression in terminals?**

Thank you for your indication. As you suggested, mCherry expression in the PP seemed to be unspecific. We tried to re-evaluate the mCherry expression in the PP. Since 30 μm section was not enough to distinguish the signals, 4 μm of paraffin-embedded pituitary section was prepared. FIHC for OT was performed in the section. Although it is still

possible that the mCherry signals were non-specific, we hope the figure is now improved for the readers. In addition, mCherry signals that were observed in the posterior pituitary were not found in the anterior and middle lobe of the pituitary. Abundant mCherry signals (confirmed by FISH for OT) were observed in the brainstem and spinal cord (Supple. Fig. 5G), suggesting that mCherry might be expressed in the posterior pituitary, the terminals of OT neurons, as well. Please see our revised Supple. Fig. 2A.

10. It is great that the unexpected significant effect in WT controls is reported. How would you explain this effect, and how does it not invalidate the findings in the transgenic rat line?

Thank you very much for your indication. CNO contains a similar structure of clozapine, an antipsychotic drug that may lead to an up-regulation of TPH (Donohoe *et al.*, *J Neurosci Res.* 2008). Although the concentration of CNO used in our study is widely used for chemogenetic activation, and should not have major off-target effects, it is possible that it could induce the small but significant increase in TPH-ir neurons we observed in WT rats after CNO injection. In addition, several kinds of physical stress can increase the expression of TPH (Gardner *et al.*, *Neuroscience*, 2009). Of note, however, we did not see any significant increase of TPH-ir + Fos-ir neurons in the DR of WT rats (Supple. Fig. 4G). We have added these discussions in the main text. Please see the Results 4 *OT activated serotonergic neurons in the DR and inhibitory interneurons in the spinal dorsal horn* on pages 7 and 8.

11. Briefly describe (as you did for Fos and PAX2) what tyrosin-hydroxylase and tryptophan hydroxylase are a measure for. Also, why is in Supp. Fig. 3 LC only TH-ir shown, and in DR TPH-ir, while in Supp. Fig. 4 TPH-ir in LC and TH-ir in DR is shown?

Thank you for your indication. Tyrosine hydroxylase (TH) and tryptophan hydroxylase (TPH) are the rate-limiting enzyme in the biosynthesis of noradrenaline and serotonin (5-hydroxytryptamine; 5-HT), respectively (Chen *et al. Biochem Pharmacol.*, 2013; Nagatsu *et al. J Biol Chem.*, 1964). TH is expressed in noradrenergic neurons in the LC, and TPH2 is expressed in serotonergic neurons localized within the dorsal and median raphe nuclei of the CNS. We analyzed TPH2-ir and TH-ir neurons to evaluate the expression of serotonergic and noradrenergic neurons, respectively. We have revised the manuscript accordingly. We are very sorry for our mistakes regarding the supplementary figures. As you indicated, TH-ir in the LC and TPH-ir in the DR is correct. We have modified them. Please see in lines 176-182 on page 7. Please also see our revised Supple. Fig. 5.

12. Along the same lines, why is there a representative image of TH and Fos colocalisation of only DR, and only a counting and ISH image in LC?

Thank you for your suggestion. Due to the limited space, all images could not be included in the main figure. However, we have added individual images in the supplementary figure. Please see Supple. Fig. 3.

13. How do you explain that von Frey and hot plate test (mechanical vs thermal pain) have their only significant anti-nociceptive effect at different timepoints (30 vs 60 min)? Please discuss this in the manuscript.

Thank you very much for the important indication. Although we could not answer to your question reasonably, our speculation is described below. Previous study has revealed that both mechanical and thermal nociceptive thresholds were increased at the same timepoint (within 10 minutes after OT administrations). Due to its short half-life, the results of the effects of OT might be different to ours. Serum OT in our transgenic rat was significantly increased at 30 minutes and kept high concentration until at least 180 min after CNO injection. This unique pattern of serum OT concentration (i.e., more likely continuous injection rather than single injection) could be one of the reasons for the discrepancy. Thermal and mechanical sensations are transmitted by A δ and C fiber. C fiber is thinner than A δ fiber. Among the primary afferent nerves, OT may first block the nerve that transmits mechanical stimuli, then block the nerve that transmits thermal sensation. Eliava *et al.* (*Neuron*, 2016) have used optogenetics and chemogenetics to selectively activate OT neurons and evaluated mechanical and thermal nociceptive thresholds. Since the methodology is different, the results cannot be compared to our data. As we recognize these are too speculative and not convincing enough, we did not add these discussions into main text.

14. While the difference between OTR a i.p. + vehicle (blue) and OTR a i.p. + i.t. (yellow) in the inflammatory pain model is not significant based on your figures (as compared in Fig. 6 H,I), there seems to be a clear difference between the two at 20-40 minutes? How do you explain this?

Thank you for your indication. Indeed, there were significant differences. We have added them in the figures. Please see our revised Fig. 6 and Fig. 7.

15. In the hot plate/von Frey test ablation by OTR antagonist completely alleviates the effect back to the baseline behaviour observed after saline injection. Why is this not the case in the inflammatory pain model in the first 20 minutes (Fig. 6 E, 6, black vs blue and yellow line)?

Thank you for your suggestion. According to the results of the Seltzer model, neuronal OT-ergic pathway plays a greater role in the inhibition of mechanical/heat transmissions, than the humoral OT pathway does. In the formalin test model, however, the effects of CNO on licking time were not affected by i.t. injection of OTR antagonist, but was ablated by i.p. injection of OTR antagonist. The results demonstrated that humoral OT might play a greater role in the reduction of spontaneous nociceptive behaviors and inflammations. We have revised the manuscript to describe these clearer. Please see the Results 6 and 7 on pages 9 and 10.

16. If animals are briefly anesthetized for formalin injection (probably with a variance between animals regarding the “deepness” of anesthesia?) how can you start behavioural readout (e.g. licking) at timepoint 0?

Thank you for your indication. In the present study, timepoint 0 was defined as the commencement of licking their formalin-injected limb. As you suggested, the deepness of anesthesia differed in each rat, however the time differences from formalin injection to timepoint 0 was less than 1 minute among all rats. We have added the definition in the Methods. Please see *Formalin test* on pages 20 and 21.

17. Please mention that OT does not cross the BBB, therefore making the differentiation into neuronal/humoral after i.p. and i.t. injection possible.

Thank you for your important indication. We have added those in the Results. Please see in lines 270-271 on page 9.

18. Please also provide the number of counted cells and not only their percentage (dpPVN vs mPVN). Particularly for dpPVN counting of %Fos colocalization seems to be based on very little cells?

Thank you for your indication. We have revised the Figure. Please see our revised Fig. 2E.

19. How/why did you define the first and second phase of licking behaviour etc. as 0-10 and 10-60 minutes?

Thank you for your indication. Pain induced by formalin in rodents has two phases which reflect different pathological processes. After the formalin injection, animals show early or acute painful responses (0-7 min) which imitate the direct activation of nociceptors, then, attenuation or quiescent of nociceptive responses in an interphase is observed, followed by a long-lasting period of nociceptive behaviors which might last for more than

45 minutes. According to the previous study, (Yamamoto *et al.*, *British Journal of Pharmacology*, 2002), we defined the 1st and 2nd phase as 0-10 and 10-60 minutes, respectively. We defined the phase 1 and 2 as 0-10 and 10-60 minutes, respectively. We have added the explanation in the Methods. Please see *Formalin test* on pages 20 and 21.

20. In figure 1I, the legend for PVN is missing (in accordance to Fig. 1F)

We are very sorry for our mistakes. We have revised the Figure. Please see our revised Fig. 1F.

21. The conclusion “OT may be involved in the remodeling of the injured spinal cord” should be explained.

Thank you for your suggestion. According to the results of the RNAseq, many GO terms that were involved in spinal remodeling were enriched after CNO injection. Indeed, Gumus *et al.*, have reported that OT had protective effects on nerve recovery in the sciatic nerve damage model in rats (Gumus *et al.*, *J Orthop Surg Res.*, 2015). In addition, OT administration resulted in accelerating the recovery from sciatic nerve injury in rats (Gutierrez *et al.*, *Anesthesiology*, 2013). We have revised the Discussion. Please see in lines 412-417 on page 13.

22. p18 I595 Mice?

Thanks for your comment. Rats but not mice. The animal species used in the reference was also “rats”.

23. supp. fig. 7D Do you have an image with rats in the same position? the CNO animal is angled which makes comparison difficult

Thank you for your suggestion. We have replaced the image. Please see our revised Supple. Fig. 8F.

24. Fig. 7 A-C Please indicate injection site of saline (it’s clear for CNO and formalin, but for saline it would be nice to have “saline s.c. (in paw)” added to the legend)

Thank you for your suggestion. We indicated injection site clearer in the Figure. Please see our revised Fig. 8.

25. Spelling/grammar

The manuscript should be checked for correct use of articles (a/an, the) and

punctuation (too many/little commas). Examples:

- p3 l46 “THE descending pain inhibitory system”
- p3 l51 “of AN OT receptor antagonist”
- p3 l52 “altering THE hypothalamus-pituitary-adrenal axis”
- p4 l82 “involved in pain modulation”

Please change direct referring to figures to be consistently: “in the figure (Fig. X)” vs “in figure X”

Typos and rephrasing

- p3 l48ff the sentence should be rewritten to be more clearly
- p4 l64 “neuronal populationS”
- p4 l66 replace “Whilst” with e.g. “On the other hand” or even “Meanwhile”
- p4 l75ff make sentence clearer with structuring, e.g. “on the one hand”/“on the other hand” or “firstly”/“secondly”
- p4 l76 “in the periphery”
- p4 l80 “stress-conditions / stressful conditions”
- p4 l91f rephrase “via centrally and peripherally using the transgenic rats.” to clarify
- p5 l99f What is the meaning of this sentence? Was the line “established” after the following described verification of expression?
- p5 l125f Double-negative. “Consistently, neither serum OT nor VP remained unchanged after the s.c. injection of CNO in WT rats” -> means that both changed, which is not the case. “Consistently, both serum OT and VP remained unchanged after the s.c. injection of CNO in WT rats” or “Consistently, neither serum OT nor VP changed after the s.c. injection of CNO in WT rats”
- p5 l141 replace “Whilst”, see above
- p6 l145 “OT altered anti-nociceptive behaviour via descending pain inhibitory system” should be rephrased. What is “anti-nociceptive behaviour”? It is being used synonymously with “nociceptive behaviour” throughout the manuscript.
- p6 l146 “were carried out”
- S Figure 3 heading “did not ... neither ... nor” is over-negation and grammatically incorrect
- p6 l158 “were arisen” replace by “did arise”
- p7 l169 add comma for clarification “(TPH)-ir neurons, Fos-ir neurons, and TPH-ir neurons co-expressed with Fos-ir...”
- p7 l188 “... we speculated there to be direct...”
- Supp. Fig.4D “PVN” split into second line
- p7 l199 remove point after quality
- p8 l216 “are rapidly developed, and continue for a substantial amount of time”

- p8 l217 “were promptly”
- p8 l219 “to exclude unwanted effects of surgical intervention, the experiment was performed”
- p8 l225 missing substantive “effects of OT on neuropathic pain were ascribed to the neuronal or humoral targets of OT”
- l230 double negation, see above
- l231 “both” confusing in this context, better “whilst combined i.p. plus i.t. injection”
- Fig 5. legend l924f “Saline or CNO after pretreatment with either an OTR antagonist or vehicle...”
- p8 l245 “Total licking time”
- p8 l246 “decreased in animals pretreated with CNO compared to SaLine”
- p8 l245f and p9 l251 “analysed every 5 min” would be better “analysed in 5 min intervals or bins”
- p9 l252 “vehicle”
- p9 l285 please rephrase “crucial confrontations”
- p9 l287f double negation, see above
- p10 l304 “elicit” wrong word in this context
- p10 l314 “in the lateral wing of the DR which is a stress-sensitive region”
- p11 l350 “These results suggest that...”
- p12 l374 “mast cells by modifying”
- p16 l 511 “brains and pituitaries”

Thank you very much for indications. Your careful and rigorous review helped us to improve the manuscript for better readability. We have amended these indications accordingly. In addition, the manuscript was edited by a native English user. Please see and re-check our manuscript.

Reviewer #2 (Remarks to the Author):

- 1. A confusion exist in this paper regarding the terminology. Anti-nociception refers to a decrease (or abolition) of normal nociception (Figure 2) while anti-hyperalgesia refers to a decrease in hyperalgesia existing in a pain model (Figure 5). Authors should carefully review this. The definitions provided by the IASP might help (See DOI: 10.1097/j.pain.0000000000001939).**

Thank you for your indications. We have carefully revised the term “anti-nociception” and “anti-hyperalgesia” throughout the manuscript.

- 2. Important references are missing, such as Eliava et al., Neuron, 2016. Here, authors used similar approach (DREADD and optogenetics) to decipher one of the putative endogenous OT analgesic pathways. A new (apparently not yet peer reviewed) study from the same labs is now online (biorxiv) regarding a new OT pathway toward periaqueductal grey. These make the statement at the end of introduction, p4, incorrect “Direct effects of endogenous OT on pain pathways, however, have not been clarified”. While it does not affect the interest of Nishimura et al manuscript, I strongly suggest authors to modify this statement. It is particularly important for the authors to compare their results to other studies involving endogenous oxytocin. This can also be used to enrich the discussion part when approaching the role of OT in spinal cord.**

Thank you very much for your constructive comments. We are very sorry for missing important references which are now included in the main text. We have revised the Introduction according to your suggestion. Please see our revised Introduction on pages 4 and 5.

- 3. Page 6, authors say “DR and LC, the nuclei that are involved in the descending pain inhibitory system”. I agree those are part of the descending controls. However, many other structures are involved, even if only considering structures in which OT has been suggested to act, such as PAG, RVM, PB, Amy; this has been reviewed several times (e.g. DOI: 10.1007/7854_2017_14). Thus, authors should thus modify the sentence for “DR and LC, two nuclei that are involved in the descending pain inhibitory system”. The discussion should also be slightly reviewed considering it, as for now a naive reader could understand that only DR and LC are involved in descending pain controls, while it is wrong.**

Thank you very much for your comment. We have amended the manuscript to avoid the misleading for the readers. Please see in lines 175-176 on page 7. Please also see our revised Discussion.

- 4. Figure 5. Do authors measured the effect of DREADD activation of OT neurons in Sham animals as well? I failed to find this information. Please include the results, as it is a mandatory control for any neuropathic pain model.**

Thank you very much for your suggestion. We performed additional experiment as we failed to include the results in the original manuscript. Sham operated animals were administered Saline or CNO. The results were similar to those of naïve animals. We have added this information in the main text and supplementary figure. Please see in lines 257-259 on page 9. Please also see Supple. Fig. 7C and 7D.

- 5. Formalin test: as far as I know, formalin test is used as a nociceptive test. This is not a classical inflammation-induced pain sensitization, as are carrageenan or CFA models. Formalin injection induces a short lasting “spontaneous” nociceptive behavior, while carrageenan/CFA injections induce an inflammation-related hyperalgesia. Authors should consider it while revising their manuscript, in both description of the text and discussion.**

Thank you for your suggestion. We have thoroughly revised our manuscript to distinguish Formalin test model as to evaluate “spontaneous nociceptive behaviors”.

- 6. In discussion, authors mention the practical use of OT in clinics. However, three main parameters are limiting its use there: first, its very short half life (5-15min, depending on the compartment), second, its very poor BBB passage, third, its ability to bind V1a, V1b and V2 receptors, with foreseen dangerous side effects. Therefore, chemists are now looking for more specific OTR agonists with longer half life. To my knowledge, only one is currently under investigation for pain (doi: 10.1038/s41598-020-59929-w). This might be mentioned in the discussion.**

We are grateful to your important indications. We have revised the Discussion accordingly. Please see in lines 425-433 on page 13.

- 7. While the discussion is interesting, and I really appreciate that author don't overestimate their results, they should add a paragraph regarding the new rat line: it is a major achievement in the field, and highlighting the specificity of expression and the further use for of this tool might be of interest. I believe it is a strong tool, which should not be overlooked.**

Thank you for your suggestion. We have revised the Discussion, including the new paragraph of generating the novel transgenic rat line which was one of the major achievements in the field. Please see in lines 459-473 on page 14.

8. Please mention in the text and figure legend the concentrations / quantities used for all molecules.

Thank you for your indication. We have revised them accordingly. Concentrations and quantities are now included in the main text and figures.

9. The OTR antagonist used (L368) (please mention it in the core of the manuscript, not only in methods, for clarity purpose) is known to pass through the BBB. 12ul at 1ug/ul was injected i.t. but how much i.p.? Is it possible that the difference observed between i.t. and i.p. injections is due to a difference in doses?

Thank you for your suggestion. OTR antagonists used in the present study are different. We have revised the manuscript as we failed to describe the accurate information in the original version. Rats were either treated with an intraperitoneal (i.p.) injection of OTR antagonist (L-371,257, dissolved in dimethyl sulfoxide (DMSO) [10 mg kg⁻¹], which has poor ability to penetrate the blood brain barrier (BBB)), intrathecal (i.t.) injection of OTR antagonist (Atosiban, dissolved in saline [1 µg µL⁻¹, 12 µg per rat]) or both (n=9 rats, each). Since OT itself does not cross the BBB, the role of neuronal/humoral OT could be differentiated by using these antagonists. The dose of i.t. OTR antagonist was very small that the effects on periphery was negligible. We have revised the manuscript, including them in the core of the manuscript. Please see in lines 266-272 on page 9.

10. Figure S2C, It would be nice if authors additionally provide a low magnitude picture of the posterior pituitary.

Thank you for your indication. Since mCherry expression in the PP seemed to be unspecific, we tried to re-evaluate the mCherry expression in the PP. Since 30 µm section was not enough to distinguish the signals, 4 µm of paraffin-embedded pituitary section was prepared. FIHC for OT was performed in the section. Although it is still possible that the mCherry signals were non-specific, we hope the figure is now improved for the readers. In addition, mCherry signals that were observed in the posterior pituitary were not found in the anterior and middle lobe of the pituitary. Abundant mCherry signals (confirmed by FIHC for OT) were observed in the brainstem and spinal cord (Supple. Fig. 5G), suggesting that mCherry might be expressed in the posterior pituitary, the terminals of OT neurons, as well. We have revised the supplementary figure as you suggested. Please see our revised Supple. Fig. 2A.

- 11. Line 35, “mPVN” refers to medial paraventricular nucleus. It should be define, as otherwise one might believe it is magnocellular PVN.**

Thank you for your indication. mPVN refers to “magnocellular PVN”. We have defined them in the manuscript.

- 12. Line 96, I guess “BAC” should be defined. Same for “FIHC” line 116**

Thank you for your suggestion. We have defined BAC in the manuscript. FIHC was also defined in the manuscript. Please see in lines 108 and 114 on page 5.

- 13. Line 99, figure appeal seems to be wrong, 1B should be 1B-C. There are other small mistakes of that order in the manuscript that should be carefully reviewed.**

Thank you very much for your indications. We have carefully revised the manuscript thoroughly.

- 14. Line 132, the official nomenclature for oxytocin gene is OXT. For vasopressin, this is AVP.**

Thank you for your indication. We have amended *oxytocin* and *vasopressin* genes as *OXT* and *AVP*. Please see in lines 149 and 159 on page 6.

- 15. Line 141, VP must be defined**

Thank you for your indication. VP had been defined in the Result. Please see in line 114 on page 5.

- 16. Line 204, reference in wrong format (Haring et al., 2018)**

Thank you for your indication. We have amended the format of the reference.

- 17. Page 8, please briefly describe Seltzer model of neuropathic pain in the core of the text, for clarity purpose.**

Thank you for your indication. We have added a brief description of Seltzer model in the Result. Please see in lines 252-253 on page 9.

- 18. Knee inflammation: please mention in the text (not only mat and meth) the model used.**

Thank you for your indication. We have added detailed descriptions of Carrageenan knee arthritis model in the main text as well. Please see in lines 304-312 on page 10.

19. Some typo mistakes are present within the manuscript (e.g. food should be foot in page 9)

Thanks for your indications. We have revised the typos and grammars. In addition, the manuscript has been checked by a native English user. Please also see our answer No. 25 to the reviewer #1.

Reviewer #3 (Remarks to the Author):

- 1. The major problem with the text (aside from the English, which will need to be further edited) is that the authors never really set up what was unclear about the role of oxytocin in pain. They give a reasonable introduction to oxytocin anatomy, and pain anatomy, and then simply cite a few papers suggesting that oxytocin is “involved in pain modulation and exerts anti-nociceptive effects”. But the situation is rather more complex than this, and one complication that is mentioned absolutely nowhere in the paper is the findings of Schorscher-Petcu and colleagues in 2010 suggesting that oxytocin analgesia is entirely due to activation of vasopressin-1A receptors. This paper obviously comes to a very different conclusion. Is it a rat/mouse difference? I would have expected much of the discussion to centre around this particular discrepancy, but that paper isn’t even mentioned.**

Thank you very much for your important suggestion. As you indicated, V1aR plays important roles for analgesic effects of OT. A completely specific OT receptor (OTR) antagonist has proved elusive; therefore, it has been nearly impossible to confidently determine which receptor is responsible for the analgesic actions of OT. In this work, we have used the transgenic rats to provide compelling data to support the hypothesis that the analgesic effect of OT is mediated predominantly via OTR, although we still cannot completely rule out the possibility of a role for V1aR. We have revised the Introduction. We have also revised the Discussion accordingly. Please see our revised Introduction and Discussion.

- 2. I’m somewhat worried about the behavioural pain testing, which based on the methods section is not a core competency of this group. The hot-plate test description is almost completely incomprehensible, as the dependent measure is described as “withdrawal”. What does that mean? Accepted dependent measures on the hot-plate test (which probably should not have been performed twice, because rodents learn escape behaviours with repeated exposure) include hind paw licking, lifting, shaking/fluttering. Were these all measured? For the von Frey test, the psychophysical approach was not stated. Were filaments applied in ascending order? Two stimuli per second is WAY more frequent than von Frey filaments should be applied; this kind of paradigm will produce temporal summation. This all being said, the findings are so robust that even though I think these assays were performed poorly, the findings are probably accurate nonetheless. But clarifications are needed to the methods.**

Thank you for your suggestion. “Withdrawal” for hot-plate test includes licking, lifting, or shaking the hind paw, or jump. von Frey test was performed using ascending method. Time interval between each stimulus was not 0.5 sec, but was over 3 sec. We tested hot-plate only once, not twice. We apologize for our mistakes. The paradigm which we used may produce temporal summation as you suggested. We will carry out behavioral pain testing more carefully in further studies. We have revised the Method. Please see *von Frey and hot plate tests* on page 20.

- 3. I have absolutely no idea why RNAseq was performed here. In my mind this is a classic example of performing a technique because it exists instead of because it answers a question you need answered. What conclusion do the authors come to about the hundreds of regulated genes? That oxytocin modulates pain both directly and indirectly, that inflammatory/immune response might be involved, and that “many” of the genes are expressed in GABAergic neurons. None of these hypotheses was followed up directly.**

Thank you for your suggestion. We speculated that functional gene expression pathways related to anti-nociception might be changed after OT activation, since abundant innervations from OT^{PVN} neurons were observed in the spinal dorsal horn. We therefore employed RNAseq to investigate this hypothesis in an unbiased manner, rather than using a targeted candidate approach. Although it is difficult to conclude, we presumed the altered genes might be relevant to pain transmission in the spinal dorsal horn from the results. This is the rationale why we used RNAseq in the present study. We have added the explanation for conducting RNAseq in the manuscript. Please see the Results 5 *OT induced differential gene expression in GABA-ergic neurons of the dorsal horn* on page 8.

- 4. The figures are dense, not high-enough resolution, and do not conform to the increasingly universal expectation that individual data points will be shown.**

Thank you for your indication. Figures have been revised, not to be too dense. We were sorry for our inadequate resolution of the figures since we had to submit the manuscript as one PDF file including the figures. The original figures, which would be submitted if the manuscript will be accepted, are high enough resolution. We will upload separate PDF file of the figures in our revised version. In addition, individual data points were shown to conform to the universal expectation as much as possible.

- 5. If “the dorsal part of the PVN” really needs to be abbreviated, it should be “dPVN”, not “dpPVN”.**

Thank you for your suggestion. We have modified “dpPVN” into “dPVN” in all cases.

- 6. Why were two completely different OTR antagonists used in the i.p. versus i.t. injections? There might be a reason related to permeability or solubility, but this should be defended.**

Thank you very much for your suggestion. We used different OTR antagonists for differentiating the role of neuronal/humoral OT, especially because of their permeability to the BBB. The dose of i.t. OTR (much cheaper) was very small that the effect on peripheral OT might be negligible. In addition, since i.p. OTR (much expensive) used in this study does not cross the BBB, it is possible to differentiate the role of neuronal/humoral OT by applying these different OTR antagonists. We have added the information of OTR antagonists in the manuscript. Please see in lines 266-272 on page 9.

- 7. The mast cell hypothesis would be strengthened greatly by the use of a mast cell stabilizer, instead of trying to rule out the alternative via HPA axis gene expression.**

Thank you very much for your suggestion. We performed additional experiment. Saline, CNO (1 mg kg^{-1}), or Disodium cromoglicate (DSCG, 50 mg kg^{-1}), commonly used as mast cell stabilizer, was s.c. administered at 30 min prior to the experiment. Then, 5% formalin ($100 \mu\text{L}$) was s.c. injected into the right hind paw (formalin test). Strikingly, the effects of DSCG were comparable to CNO. Licking time was significantly decreased. In addition, foot pad thickness after the 5% formalin injection was significantly attenuated. Thanks to you, the mast cell hypothesis is now much strengthened. We could not have done such experiment without your opinion. We have added these results in the supplementary figure and manuscript. We have also explored HPA axis under pathological condition since another reviewer suggested to do so. Although endogenous OT did not affect HPA axis under naïve condition, OT possibly affects HPA axis under pathological condition. We have revised the manuscript by adding these additional results. Please see in lines 329-334 and 349-361 on page 11. Please also see our revised Fig. 9H-K, Supple. Fig. 9, and Supple. Fig. 10.

- 8. Why do Figures 5 and 6 variously say “OTR antagonist” versus “OXTR antagonist”?**

We are very sorry for our mistakes. We have revised the figures.

- 9. It should be made clearer to the reader that in fig. 5F-G and Fig. 6G-I that CNO is on-board as well as the other drugs.**

Thank you very much for your suggestion. We have revised the Figure clearer that CNO was on-board. Please see Fig. 6 and 7.

Reviewers' comments:

Reviewer #1 (Remarks to the Author):

The manuscript has been significantly improved. The authors have addressed all my concerns, thus I recommend publication of this important study.

Reviewer #2 (Remarks to the Author):

The authors have addressed most of my remarks and comments. I appreciate the improved manuscript and would like to suggest a few more corrections / additions to the manuscript, particularly in discussion.

1. As a part of the discussion, I suggest authors to do an extensive literature of opto- and chemo-genetics performed up to now on OT neurons. They clearly improved this bibliographic part in their revised manuscript but some are still lacking. Importantly, by doing so they can expose what was done with which mean (AAV in rats, transgenic lines in mice), compare the technical approaches and propose to overcome the AAV-induced difficulties with a new rat line. By doing this in a clear and positive way, they will be able to explain to the reader the interest of this line and how complimentary it is with viral approaches. They will also be able to expand the scope to their study to other fields of research not directly related to pain, such as social interactions. It will only be beneficial. Among the papers to referee, I can think about <https://doi.org/10.1101/2022.02.23.481531>; doi: 10.7554/eLife.73421; doi: 10.1126/science.aan4994; doi: 10.1016/j.neuron.2017.06.003.

2. Regarding the Sham + CNO experiment (Fig. S7) that I congratulate the authors for, it seems that CNO induces a transient analgesia, as in naïve animals (Fig 3). Authors should discuss this in the discussion, as it is slightly different to what was previously published in rats when manipulating OT neurons opto / chemogenetically. Probably, the difference comes from the targeting of a larger population, not linked to a discrete neuronal circuit to spinal cord (doi: 10.1016/j.neuron.2016.01.041.) or PAG (<https://doi.org/10.1101/2022.02.23.481531>).

3. While authors have added the drug concentrations in the figure legends, I suggest them to include it in the figures themselves to improve visibility.

4. L371 OTR antagonist. Authors claim is has poor ability to cross the BBB. As far as I remember, many people use it for its ability to cross the BBB. Authors should probably remove the "which has poor ability to penetrate the BBB" sentence.

5. In the same purpose, for figure 6, (and others) authors should explicitly mention which OTR antagonist they injected in the figure legend, instead of writing "OTR antagonist". Please explicitly write L371, Atosiban or L371+Atosiban in all figures mentioning OTR antagonist. Same for vehicle. Also, authors must be aware that atosiban is NOT an antagonist, but a biased agonist of OTR-Gi pathway – they should discuss this point in the discussion.

Reviewer #3 (Remarks to the Author):

The authors were extremely responsive to my suggestions, including performing a new experiment, and the manuscript is much stronger now. I still have some issues with the behavioural testing, but I don't believe this compromises the conclusions. I would insist that the authors acknowledge in the

methods section that the application of 5 von Frey fibers in 3 seconds may induce temporal summation, and also that the hot plate behaviours quantified not be characterized as "withdrawal", but rather as "nocifensive behaviors".

We are grateful to the reviewers and editor for their rigorous and constructive review of our manuscript. Their careful review helped us to improve our manuscript for better readability. Our answers to the individual comments are enclosed.

Reviewer #1 (Remarks to the Author):

1. **The manuscript has been significantly improved. The authors have addressed all my concerns, thus I recommend publication of this important study.**

Thank you very much for your careful review of our manuscript.

Reviewer #2 (Remarks to the Author):

1. **As a part of the discussion, I suggest authors to do an extensive literature of opto- and chemo-genetics performed up to now on OT neurons. They clearly improved this bibliographic part in their revised manuscript but some are still lacking. Importantly, by doing so they can expose what was done with which mean (AAV in rats, transgenic lines in mice), compare the technical approaches and propose to overcome the AAV-induced difficulties with a new rat line. By doing this in a clear and positive way, they will be able to explain to the reader the interest of this line and how complimentary it is with viral approaches. They will also be able to expand the scope to their study to other fields of research not directly related to pain, such as social interactions. It will only be beneficial. Among the papers to refer, I can think about <https://doi.org/10.1101/2022.02.23.481531>; doi: 10.7554/eLife.73421; doi: 10.1126/science.aan4994; doi: 10.1016/j.neuron.2017.06.003.**

Thank you very much for your indications. We discussed and added an extensive literature review to emphasize the uniqueness and advantage of our transgenic rats. Please see in lines 483-493 on page 15.

2. **Regarding the Sham + CNO experiment (Fig. S7) that I congratulate the authors for, it seems that CNO induces a transient analgesia, as in naïve animals (Fig 3). Authors should discuss this in the discussion, as it is slightly different to what was previously published in rats when manipulating OT neurons opto / chemogenetically. Probably, the difference comes from the targeting of a larger population, not linked to a discrete neuronal circuit to spinal cord (doi: 10.1016/j.neuron.2016.01.041.) or PAG (<https://doi.org/10.1101/2022.02.23.481531>).**

Thank you very much for your constructive comments. We have added the discussion, accordingly. Please see in lines 391-396 on page 12.

3. **While authors have added the drug concentrations in the figure legends, I suggest them to include it in the figures themselves to improve visibility.**

Thank you very much for your comment. We have included the drug concentrations in the figures to improve visibility. Please see Fig. 6 and 7.

4. **L371 OTR antagonist. Authors claim it has poor ability to cross the BBB. As far as I remember, many people use it for its ability to cross the BBB. Authors should probably remove the “which has poor ability to penetrate the BBB” sentence.**

Thank you very much for your suggestion. We have deleted the sentence.

5. **In the same purpose, for figure 6, (and others) authors should explicitly mention which OTR antagonist they injected in the figure legend, instead of writing “OTR antagonist”. Please explicitly write L371, Atosiban or L371+Atosiban in all figures mentioning OTR antagonist. Same for vehicle. Also, authors must be aware that atosiban is NOT an antagonist, but a biased agonist of OTR-Gi pathway – they should discuss this point in the discussion.**

Thank you for your suggestion. We have revised the figures and figure legends. We have also added the information of atosiban in the Discussion. Please see in lines 429-432 on page 13. Please also see Fig. 6 and 7.

Reviewer #3 (Remarks to the Author):

1. **The authors were extremely responsive to my suggestions, including performing a new experiment, and the manuscript is much stronger now. I still have some issues with the behavioural testing, but I don't believe this compromises the conclusions. I would insist that the authors acknowledge in the methods section that the application of 5 von Frey fibers in 3 seconds may induce temporal summation, and also that the hot plate behaviours quantified not be characterized as "withdrawal", but rather as "nocifensive behaviors".**

Thank you very much for your suggestions. Yes, the possibility of temporal summation could not be ignored. This was our failure which should be technically refined in further studies. Thus, we described the possibility in the Methods. Please see in lines 701-702 on page 21. In addition, we have amended the terms “withdrawal” associated with hot plate test into “nocifensive behaviors” in all cases, including figures.

REVIEWERS' COMMENTS:

Reviewer #2 (Remarks to the Author):

The authors fully addressed all my comments. I strongly believe that the manuscript is now suitable for publication and will attract the attention of the community. Congratulations!